# Mature IgD$^{low/-}$ B cells maintain tolerance by promoting regulatory T cell homeostasis

Avijit Ray[1,4], Mohamed I. Khalil[1,2], Kirthi L. Pulakanti[1], Robert T. Burns[1], Cody J. Gurski[1], Sreemanti Basu[1,3,5], Demin Wang[1,3], Sridhar Rao [1] & Bonnie N. Dittel [1,3]

A number of different B cell subsets have been shown to exhibit regulatory activity using a variety of mechanisms to attenuate inflammatory diseases. Here we show, using anti-CD20-mediated partial B cell depletion in mice, that a population of mature B cells distinguishable by IgD$^{low/-}$ expression maintains tolerance by, at least in part, promoting CD4$^+$Foxp3$^+$ regulatory T cell homeostatic expansion via glucocorticoid-induced tumor necrosis factor receptor ligand, or GITRL. Cell surface phenotyping, transcriptome analysis and developmental study data show that B cells expressing IgD at a low level (BD$_L$) are a novel population of mature B cells that emerge in the spleen from the transitional-2 stage paralleling the differentiation of follicular B cells. The cell surface phenotype and regulatory function of BD$_L$ are highly suggestive that they are a new B cell subset. Human splenic and peripheral blood IgD$^{low/-}$ B cells also exhibit BD$_L$ regulatory activity, rendering them of therapeutic interest.

---

[1] Blood Research Institute, Versiti Wisconsin, Milwaukee, WI, USA. [2] Molecular Biology Department, National Research Centre (NRC), El-Buhouth Street, Dokki, Cairo, Egypt. [3] Department of Microbiology and Immunology, Medical College of Wisconsin, Milwaukee, WI, USA. [4] Present address: Oncology Discovery, AbbVie Inc., North Chicago, IL, USA. [5] Present address: AbbVie Inc., North Chicago, IL, USA. These authors contributed equally: Avijit Ray, Mohamed I. Khalil. Correspondence and requests for materials should be addressed to B.N.D. (email: bonnie.dittel@bcw.edu)

A regulatory role for B cells in controlling the severity of autoimmunity was first described by us in the mouse model of multiple sclerosis (MS), experimental autoimmune encephalomyelitis (EAE)[1]. Specifically, we showed that B10.PL mice deficient in B cells (µMT) immunized with the myelin basic protein (MBP)-immunodominant peptide $Ac_{1-11}$ were unable to recover from the signs of EAE exhibiting a chronic disease course[1]. We subsequently reproduced these findings in mice on the B10.PL background using adoptive transfer EAE and in anti-CD20-depleted mice[2,3]. Our original findings were replicated in C57BL/6µMT mice immunized with the myelin oligodendrocyte glycoprotein 35–55 peptide[4]. The later study identified B cell production of interleukin-10 (IL-10) as the mechanism by which B cells regulate the severity of EAE[4]. However, numerous studies failed to identify a distinct B cell subset that regulates via IL-10[5]. In addition, it has become clear that other B cell regulatory mechanisms exist[6,7]. The existence of B cells with regulatory activity in humans has also been demonstrated, but as in mice, a definitive phenotype has remained elusive[8].

We were the first to identify an IL-10-independent regulatory B cell mechanism operational in EAE[3]. We found that µMT and CD20 B cell-depleted mice had a significant reduction in the absolute number of CD4+Foxp3+ T regulatory cells (Tregs)[3]. B cell reconstitution of µMT mice induced Treg proliferation and maintenance resulting in resolution of EAE[3]. Tregs are essential for the maintenance of tolerance against self-antigens and when absent or depleted humans and mice quickly succumb to autoimmune manifestations[9]. We found that the ability of B cells to homeostatically expand Treg was glucocorticoid-induced tumor necrosis factor receptor ligand (GITRL)-dependent, but IL-10-independent[3]. GITR the receptor for GITRL is highly expressed by Treg and when engaged has been reported to induce Treg proliferation[10,11]. Given the importance of B cells in providing protection against pathogens, it is unlikely that all B cell subsets would have the capacity to homeostatically expand Treg, which would potentially be detrimental for pathogen clearance.

Using a partial B cell depletion strategy to enrich for B cells with regulatory activity, here we find that B cells exhibiting an IgD low (L) ($BD_L$) phenotype induce Treg expansion and promote recovery from EAE. Both genetic and developmental studies lead us to conclude that $BD_L$ are a new subset of mature B cells. Importantly, human B cells with an IgD$^{low/−}$ phenotype exhibit $BD_L$ regulatory activity by the induction of Treg proliferation. The ability to modulate Treg numbers to either suppress or enhance immune responses is a goal for the treatment of disease. Thus, the ability to harness the regulatory function of $BD_L$ is of therapeutic interest.

## Results

**Anti-CD20 IgG$_1$ B cell depletion retains regulatory activity.** In our previous studies, total B cell depletion with anti-CD20 immunoglobulin G 2a (IgG$_{2a}$) prior to EAE induction led to significantly reduced Treg numbers and the inability to recover from EAE[3], indicating that the protective B cell population was depleted. These data suggested that a specific B cell population that facilitates Treg homeostasis and EAE resolution could be identified[3]. To that end, the strategy we chose was to partially deplete B cells with anti-CD20 that contains the same antigen recognition domain, but with the IgG$_{2a}$ Fc region swapped for IgG$_1$[12]. Administration of anti-CD20 IgG$_1$ led to a significant reduction in the total number of splenic B cells that was due to 85% loss of follicular (FO) B cells, while sparing the marginal zone (MZ) subset (Fig. 1a). Representative flow cytometry plots are shown in Fig. 1b. The kinetics of anti-CD20 IgG$_1$ B cell

depletion are shown in Suppl. Figure 1a[12]. We developed a 4-color immunofluorescence strategy to visualize B cell depletion by staining for the T cell zone (CD3), B cell follicle (IgD+IgM+), and the MZ (SIGN-R1). In isotype control mice splenic architecture is shown as a distinct T cell zone surrounded by B cell follicles encircled by the MZ (Fig. 1c, left panel). Following anti-CD20 IgG$_1$ administration, interestingly, IgD expression was lost while IgM expression remained (Fig. 1c, right panel). The MZ remained intact and there was no impact on the T cell zone (Fig. 1c, right panel). In addition, there was ~30% reduction in peripheral blood B cells (Suppl. Figure 1B) and ~95% reduction in lymph node B cells (Suppl. Figure 1C). We also found that Treg numbers were not altered (Fig. 1d). Representative CD4+ T cell and Treg flow cytometry is shown in Suppl. Figure 2. When we induced EAE in the anti-CD20-treated mice they recovered similarly to wild-type (WT) and isotype control-treated mice, as compared to chronic disease in µMT mice (Fig. 1e).

We next determined whether B cells refractory to anti-CD20 IgG$_1$ depletion retained regulatory activity. Mice receiving anti-CD20 IgG$_1$-depleted B cells recovered in a similar manner as the WT controls (Fig. 1f). Subsequently, we found that the FO, but not MZ, B cell subset was responsible for recovery from EAE (Fig. 1g).

**Anti-CD20 IgG$_1$- depleted B cells are enriched for IgD$^{low/−}$.** Given that B cell regulatory activity resided within the retained FO B cell subset, they were further analyzed by flow cytometry to uncover an enriched phenotype. This was accomplished by utilizing a B cell gating strategy utilizing differential expression of IgM, CD21, and CD23[13,14]. First, B220+ B cells and differential IgM and CD21 expression was used to define three groups: IgM$^{hi}$CD21$^{hi}$ containing T2-MZP and MZ B cells, IgM$^{int}$CD21$^{int}$ containing T3 and FO B cells, and IgM$^{hi}$CD21$^{low/−}$ containing T1 and T2 B cells (Fig. 2a). We then used CD93 and CD23 expression to further delineate each of the three groups. Because B cell depletion with anti-CD20 IgG$_1$ resulted in a loss of IgD-expressing cells in the B cell follicle (Fig. 1c), we measured IgD expression on the retained FO subset and found an eight-fold enrichment in IgD$^{low/−}$ B cells or $BD_L$ (Fig. 2b). The FO IgD$^{hi}$ subset will henceforth will be referred to as FO B cells (Fig. 2b). Because B cell depletion with anti-CD20 IgG$_1$ may impact the phenotype of the remaining B cells, we repeated our staining strategy using naive mice and included additional B cell markers. First, we determined whether the reduction in cell surface expression of IgD by $BD_L$ was due to the lack of protein expression. As shown in Fig. 2c, the reduced level of cell surface IgD expression by $BD_L$, as compared to FO B cells, was paralleled by a similar reduction in intracellular IgD (Fig. 2c). MZ B cells, the control for IgD$^{low}$ expression, also had reduced intracellular IgD (Fig. 2c). For IgM, $BD_L$ expressed higher levels of cell surface IgM (Fig. 2d, top histogram), but similar levels of intracellular IgM, as compared to FO B cells (Fig. 2d). MZ B cells expressed the highest levels of both cell surface and intracellular IgM (Fig. 2d).

We then further refined the expression pattern of $BD_L$ in naive mice and found that they expressed slightly lower levels of the B cell marker B220, as compared to FO and MZ B cells (Fig. 2e). Their expression of CD20 was intermediate between FO and MZ B cells, indicating that reduced CD20 expression was not likely why they were retained (Fig. 2f). $BD_L$ expressed an intermediate level of CD23 (Fig. 2g) and expressed the lowest levels of CD21 (Fig. 2h). $BD_L$ express the highest levels of CD93 of the three subsets (Fig. 2i). These data suggest that $BD_L$ are a splenic B cell subset that to the best of our knowledge has not been previously described. These results indicate that the refined $BD_L$ phenotype is B220+IgD$^{low/−}$CD21$^{int}$CD23+CD93$^{int}$.

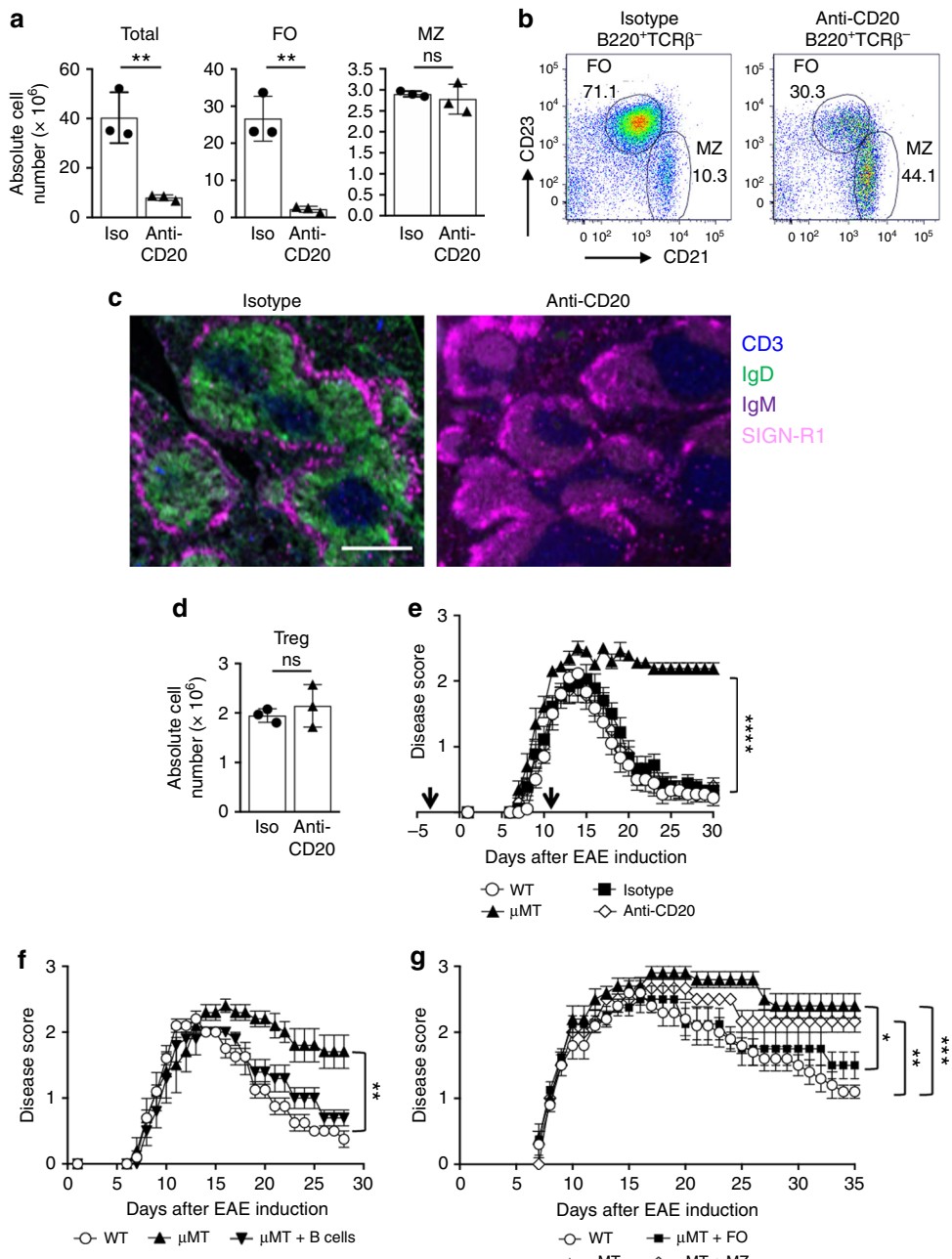

**Fig. 1** B cells after anti-CD20 (IgG₁) depletion exhibit regulatory activity. B10.PL mice were i.v. administered anti-CD20 (18B12IgG1) or its isotype control (2B8msIgG1), once (**a–d**) or twice, 14 days apart (250 μg) (**e**, black arrows). On day 14, the absolute number of total splenic B cells (B220⁺) (**a**), FO B cells (B220⁺IgM⁺CD21ⁱⁿᵗCD23⁺CD93⁻) (**a**), MZ B cells (B220⁺IgMʰⁱCD21ʰⁱCD23⁻CD93⁻) (**a**) and CD4⁺Foxp3⁺ Treg (**d**) was determined by flow cytometry. **a, d** Individual data points (mice) are shown superimposed upon the mean ± SEM. **p < 0.01. ns = not significant. **b** Representative flow cytometry dot plots with gating of splenic FO and MZ B cells shown for the isotype control (left panel) and anti-CD20 (right panel). **c** Splenic frozen sections from B10.PL mice treated with isotype control (left panel) or anti-CD20 (right panel) (250μg) were stained for CD3 (blue), IgD (green), IgM (purple), and SIGN-R1 (pink) 14 days post antibody treatment. Scale bar, 500 μM. **e** EAE was induced 3 days after the first antibody treatment by adoptive transfer of 1 × 10⁶ encephalitogenic T cells. Clinical signs of EAE were evaluated daily and the data shown are the mean ± SEM daily disease score of nine to ten mice from two independent experiments. ****p < 0.0001, μMT vs. WT, isotype or anti-CD20 for the day 30 timepoint. **f, g** B10.PL mice were i.v. administered anti-CD20 (18B12IgG1) and 14 days after antibody administration total splenic B cells (B220⁺) (**f**), FO B cells (**g**), and MZ (**g**) B cells were FACS purified and 20 (**f**) or 5 × 10⁶ (**g**) cells were adoptively transferred into sublethally irradiated B10.PLμMT mice and 3 days later EAE was induced as for **e**. WT and μMT control mice received PBS. The data shown are the mean daily disease score ± SEM of three to five mice from one representative experiment of two. **f** **p < 0.001, WT vs. μMT and μMT vs. μMT + B cells for the day 28 timepoint. **e** *p < 0.05, μMT vs. FO; **p < 0.01, WT vs. MZ; ***p < 0.001, WT vs. μMT for the day 35 timepoint. Statistical significance was determined using the unpaired t test

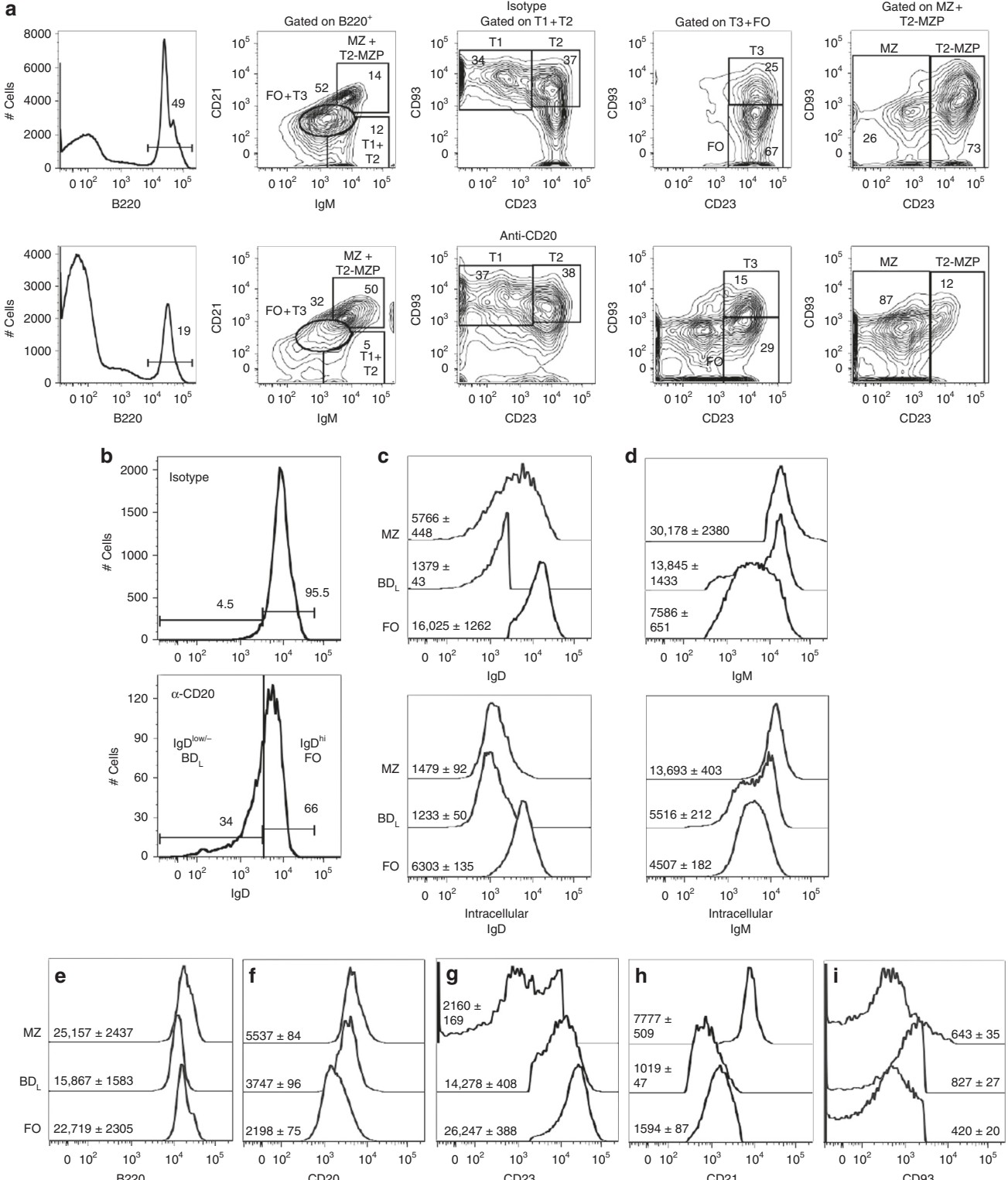

**Fig. 2** Retained B cells after anti-CD20 IgG₁ depletion are enriched for IgD^low/− expression. **a** Representative flow cytometry gating strategies in B10.PL mice are shown for splenic B220⁺ B cells differentiating T1, T2, T3, T2-MZP, FO, and MZ B cells utilizing differential expression of IgM, CD21, CD23, and CD93 14 days after administration of isotype (top row) or anti-CD20 IgG₁ (bottom row). **b** IgD expression was assessed on the FO B cell subset from **a**. **c–i** WT C57BL/6 mice were used to assess expression levels of IgD (**c**), IgM (**d**), B220 (**e**), CD20 (**f**), CD23 (**g**), CD21 (**h**), and CD93 (**i**) on MZ, BD_L, and FO B cells by flow cytometry. The mean fluorescence intensity ± SEM is shown from nine mice

**$BD_L$ regulatory activity is IL-10-independent**. We next demonstrated that $BD_L$ exhibit B cell regulatory activity by driving Treg expansion after transfer into µMT mice similarly to total B cells (Fig. 3a, b). In contrast, neither FO (Fig. 3a) nor MZ (Fig. 3b) B cell transfer led to a significant increase in Treg. To determine the lowest number of $BD_L$ required to induce maximal Treg proliferation, we performed a dose−response curve and found that $2 \times 10^6$ and $3 \times 10^6$ $BD_L$ induced similar levels of Treg expansion (Suppl. Figure 3A), but $5 \times 10^6$ $BD_L$ was required to completely restore Treg numbers (Fig. 3a, b). We confirmed that $BD_L$ induced Treg proliferation in an IL-10-independent manner (Suppl. Figure 3B). We also confirmed that $BD_L$ did not express high levels of CD5 (Suppl. Figure 3C), CD1d (Suppl. Figure 3D), or Tim1 (Suppl. Figure 3E), which are markers of IL-10-producing B cells[5].

We next tested the regulatory potential of $BD_L$ in EAE. To control for the possibility that $IgG_1$ anti-CD20 treatment modulated the function of the non-depleted cells, fluorescence-activated cell sorting (FACS)-purified splenic B cell subsets from naive mice were utilized. Adoptive transfer of $BD_L$ into µMT mice prior to EAE induction resulted in complete recovery similar to WT (Fig. 3c). In contrast, µMT control mice and those that received FO or MZ B cells did not recover (Fig. 3c). We next determined the kinetics of Treg expansion during EAE and found that in WT mice Treg numbers that started at ~$3 \times 10^6$ (Fig. 3a) were decreased in the spleen at the onset (day 7) and peak of disease (day 18), which then increased when the mice recovered from EAE (Fig. 3d). In µMT mice, Treg numbers were similar on days 0 and 7, but were decreased on days 18 and 30 (Fig. 3a vs. Fig. 3d). In µMT mice that received $BD_L$, Treg numbers were similar to WT on day 7, which were reduced on days 18 and 30 (Fig. 3d). Importantly, in µMT mice that received $BD_L$ Treg numbers were significantly elevated at every timepoint as compared to µMT mice (Fig. 3d). We also tracked the fate of conventional CD4 T cells and found that in each of the experimental conditions their numbers increased at the peak of disease and then were reduced on day 30 (Fig. 3e). As with Treg, conventional CD4 T cell numbers were significantly higher in the $BD_L$ group as compared to µMT mice (Fig. 3e). These data indicate that $BD_L$ also likely promote CD4 T cell expansion. A reduction in Treg and conventional CD4 T cells on day 7 as compared to day 0 (Fig. 3a) is consistent with the mice being sublethally irradiated for EAE induction.

To demonstrate that $BD_L$ regulatory activity is not restricted to EAE, we performed the same experiment in a contact hypersensitivity (CHS) model that does not require B cells for induction[15]. As with EAE, µMT mice exhibited a more severe disease course than WT mice that was normalized to WT levels in mice that received $BD_L$ (Fig. 3f). In contrast, when FO or MZ B cells were transferred, ear swelling was similar to the µMT control mice (Fig. 3f). Only mice that received $BD_L$ exhibited a significant increase in Treg numbers at 120 h, albeit not to WT levels, which is consistent with slightly increased ear swelling as compared to WT mice (Fig. 3f, g).

**$BD_L$ are a new B cell subset separate from FO and MZ B cells**. To determine whether $BD_L$ could be differentiated from FO and MZ B cells, we performed RNA-sequencing (RNA-seq) comparing splenic $BD_L$ to FO and MZ B cells. Using principal component (PC) analysis plotted in three-dimensional (3D), we obtained the expected result that FO and MZ B cells cluster differentially in all three dimensions (Fig. 4a). A similar result was obtained when $BD_L$ and MZ B cells were compared (Fig. 4a). To further confirm that $BD_L$ are distinct from the MZ B cell lineage, we utilized mice with a conditional deletion in *Notch2* in B cells

that lack MZ B cells[16]. We found that the percentage of $BD_L$ within the FO B cell subset (Suppl. Figure 4A) and their regulatory function were identical in *Notch2*$^{fl/fl}$ CD19Cre$^+$ and the control *Notch2*$^{fl/fl}$ CD19Cre$^-$ mice (Suppl. Figure 4B). These cumulative data provide evidence that $BD_L$ are unrelated to MZ B cells.

When we compared $BD_L$ and FO B cells, the subsets clustered similarly in PC1 and PC2 (Fig. 4a). However, in PC3 the two B cell subsets clustered separately. When we compared $BD_L$ and FO B cells, we found that 343 transcripts were significantly more highly expressed (red) in $BD_L$ and 121 transcripts were significantly less expressed (blue) (Fig. 4b). These data demonstrate that $BD_L$ express a unique transcriptome that can be used to distinguish them from FO B cells. Using gene set overrepresentation analysis, we found increased expression of genes involved in cell cycle/proliferation in $BD_L$ (Fig. 4c). Forty-four genes involved in developmental processes (Fig. 4d) and 21 genes involved in cell surface receptor signaling (Fig. 4e) were increased. Because an increase in genes regulating cell cycle/division were increased in $BD_L$ (Fig. 4c), we then measured the steady-state proliferation of splenic B cell subsets. Using Ki-67 staining proliferation was highest in the T1 and then T2 stages (Fig. 4f)[13,17,18]. As B cells matured into T2-MZP and MZ B cells proliferation decreased with the lowest level found in FO B cells (Fig. 4f). $BD_L$ exhibited identical levels of proliferation as T2-MZP (~30%) (Fig. 4f), which was significantly lower than FO B cells. Representative Ki-67 flow cytometry is shown in Suppl. Figure 5.

We then utilized flow cytometry to validate increased cell surface expression of proteins indicated in the RNA-seq. Cell surface proteins validated are CD93 (Fig. 2i), CD5 (Suppl. Figure 3c), CD9 (Suppl. Figure 6A), CD43 (Suppl. Figure 6B), and CD80 (Suppl. Figure 6C), albeit the increased expression was marginal. Cell surface proteins not validated include CD25 (Suppl. Figure 6D), F4/80 (Suppl. Figure 6E), NK1.1 (Suppl. Figure 6F), and Ly6C (Suppl. Figure 6G). We also examined additional B cell surface proteins not indicated by the RNA-seq analysis and found that $BD_L$ expressed slightly lower levels of CD38 (Suppl. Figure 6H) and CD40 (Suppl. Figure 6I) and higher levels of CD86 (Suppl. Figure 6J) as compared to FO B cells. MZ B cells were included in the analysis to further demonstrate that $BD_L$ do not share their partially activated phenotype (Suppl. Figure 6). Although both CD80 and CD86 levels were increased on $BD_L$, we previously demonstrated that neither was essential for B cells to induce Treg proliferation[3]. Although not indicated in the RNA-seq data (Fig. 4b), we showed that $BD_L$ expressed lower levels of the proinflammatory cytokine IL-6 by real-time quantitative reverse transcription PCR (Suppl. Figure 6K), which has been shown to be a B cell effector cytokine in EAE[19].

**$BD_L$ regulatory function is GITRL-dependent**. In our previous studies using total splenic B cells, we found that their regulatory activity was GITRL-dependent[3]. Consistent with this finding, we found that *Tnfs18* (GITRL) message was upregulated 2.5-fold (*p* = 0.001) in $BD_L$ in the RNA-seq analysis. When we examined the cell surface expression of GITRL, we found that all splenic B cell subsets expressed low levels of GITRL (Fig. 5a). As with the RNA-seq analysis, $BD_L$ expressed significantly higher levels of GITRL than FO B cells (Fig. 5a). Reconstitution of µMT mice with $BD_L$, but not FO B cells, resulted in a significant increase in Treg numbers (Fig. 5b). Antibody blocking of GITRL on $BD_L$ prior to transfer completely abrogated their ability to promote Treg expansion, while it had no effect on FO B cells (Fig. 5b). These findings were confirmed using *Tnfsf18*$^{-/-}$ mice, which had a significant reduction in the absolute number of Treg (Fig. 5c). In addition, *Tnfsf18*$^{-/-}$ $BD_L$ upon adoptive transfer into µMT mice

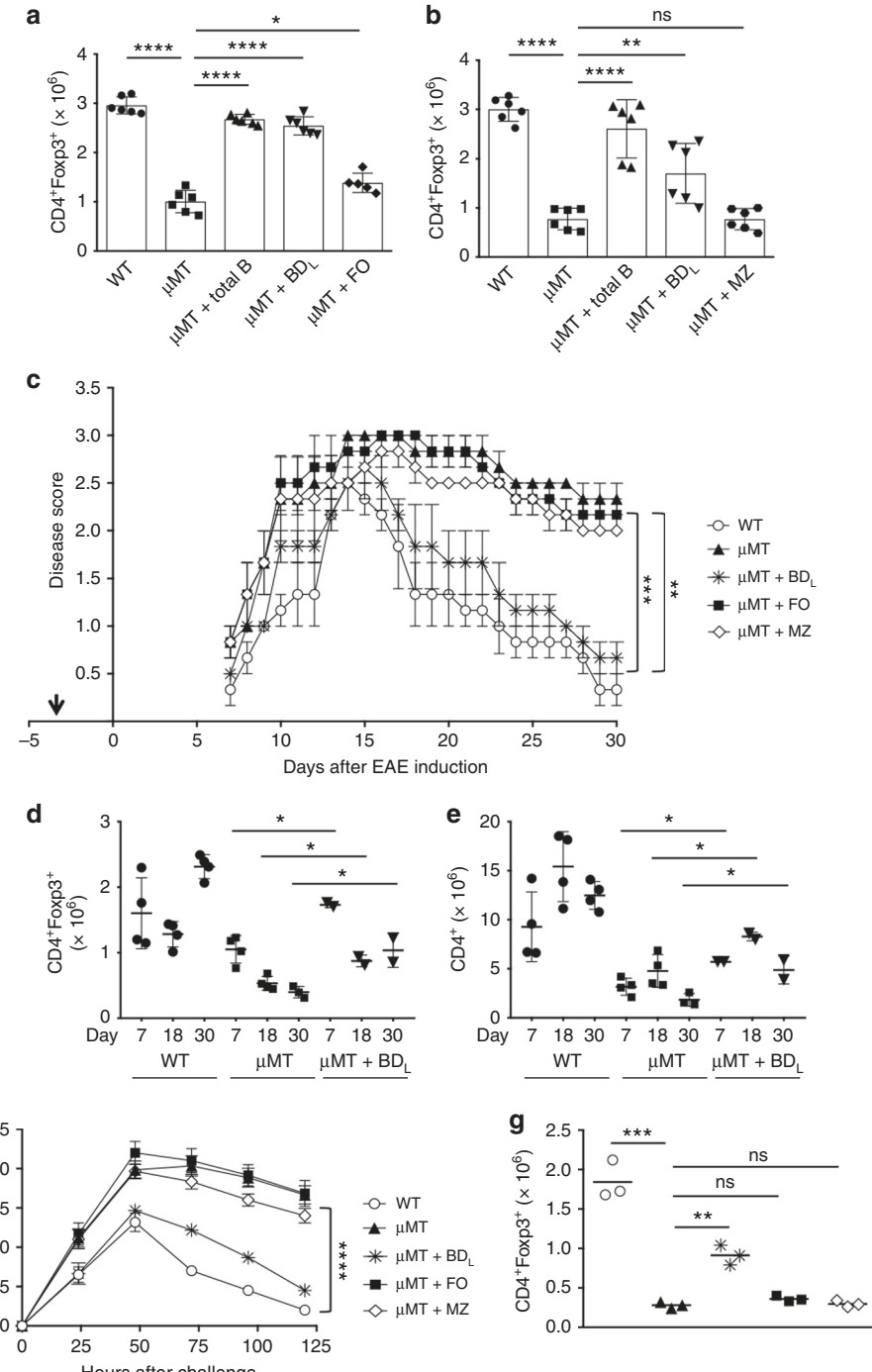

**Fig. 3** Splenic BD$_L$ exhibit regulatory activity while FO and MZ B cells do not. **a, b** B10.PLμMT (μMT) mice were treated with PBS or reconstituted with $10 \times 10^6$ FACS-purified splenic total (B220$^+$) or $5 \times 10^6$ BD$_L$ (**a, b**), FO (**a**), or MZ (**b**) B cells from anti-CD20 IgG$_1$-treated mice and the absolute number of splenic Treg was determined by flow cytometry 10 days later. B10.PL (WT) mice received PBS. Pooled data from two independent experiments are shown. Individual data points (mice) are shown superimposed upon the mean ± SEM. **a** *$p < 0.05$, μMT vs. FO; ****$p < 0.0001$, WT vs. μMT, μMT vs. Total B and μMT vs. BD$_L$. **b** **$p < 0.01$, μMT vs. BD$_L$; ****$p < 0.0001$, WT vs. μMT and μMT vs. Total B. ns = not significant. **c–e** B10.PL (WT) or B10.PLμMT (μMT) mice were sublethally irradiated and $5 \times 10^6$ BD$_L$, FO, or MZ B cells were adoptively transferred into B10.PLμMT mice (arrow) and 3 days later EAE was induced as for Fig. 1e. **c** EAE scores were assessed daily. The data shown are the mean ± SEM daily disease score of three mice from one representative experiment of three. **$p < 0.01$, WT vs. μMT, WT vs. FO, μMT vs. BD$_L$, and μMT vs. Total B; ***$p < 0.001$, WT vs. MZ for the day 30 timepoint. **d, e** The absolute number of Treg (CD4$^+$Foxp3$^+$, **d**) and conventional CD4 T cells (CD4$^+$Foxp3$^-$, **e**) was determined in the spleen by flow cytometry on days 7, 18, and 30 following EAE induction. $n = 2$–4. *$p < 0.05$. **f, g** CHS was induced in C57BL/6 WT or μMT mice treated with PBS or μMT mice that had received $5 \times 10^6$ BD$_L$, FO, or MZ B cells by painting one ear with 3% oxazolone on days 0 and 6. **f** Ear thickness was measured every 24 h for 6 days. Data shown are the mean change in ear thickness ± SEM from time 0 measured just prior to the second treatment. $n = 6$. ****$p < 0.0001$, WT vs. μMT, MZ and FO WT and μMT vs. μMT + BD$_L$ for the 120 h timepoint. **g** CD4$^+$Foxp3$^+$ Treg were quantitated in the spleen at 120 h. **$p < 0.01$, μMT vs. μMT + BD$_L$; ***$p < 0.0001$, WT vs. μMT for the 120 h timepoint. $n = 3$ mice. ns = not significant. Statistical significance was determined using the unpaired $t$ test

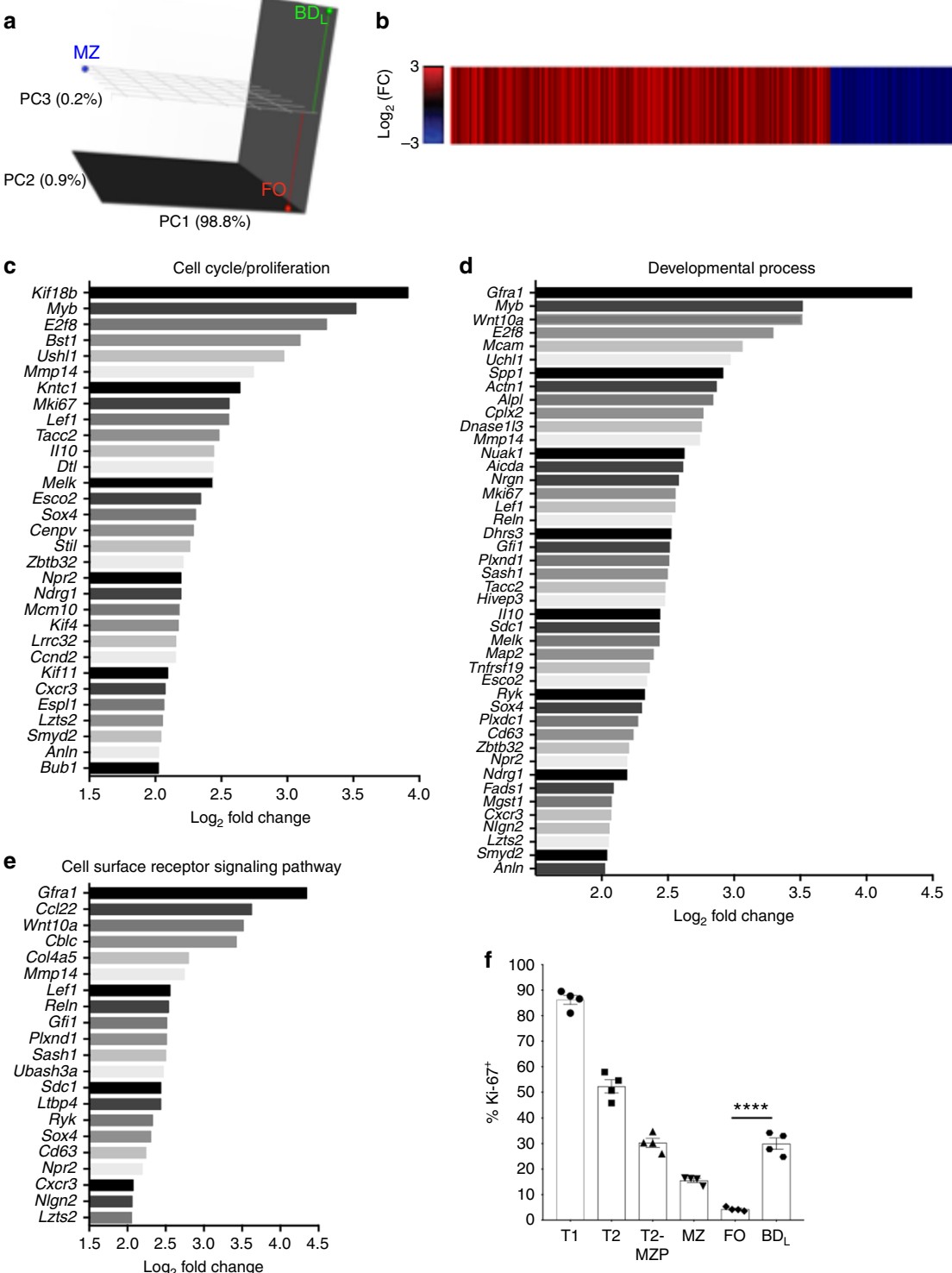

**Fig. 4** $BD_L$ undergo a high rate of proliferation. RNA-seq analysis was performed on FACS purified splenic $BD_L$, FO, and MZ B cells from B10.PL mice. **a** PC analysis was performed on the $\log_2$-transformed FPKM, with the % variance encompassed within each PC shown. The distribution of $BD_L$ (green), FO (red), and MZ (blue) B cells within the 3D plot are shown. **b** A heat map displaying the $\log_2$ fold change for genes that showed a significant difference (Benjami–Hochberg adjusted $p < 0.05$) between $BD_L$ and FO B cells are shown. Red bars show genes more highly expressed and blue reduced expression in $BD_L$ . $n = 3$. **c–e** Barplots of the most highly differentially expressed genes ($\log_2$ fold change >2) in $BD_L$ compared to FO cells within the select Gene Ontology terms cell cycle/proliferation (**c**), developmental process (**d**), and cell surface signaling pathway (**e**) are shown. **f** The percentage of splenic T1 ($B220^+IgM^{hi}CD21^{low}CD23^-CD93^+$), T2, T2-MZP ($B220^+IgM^{hi}CD21^{hi}CD23^+$), MZ, $BD_L$ and FO B cells from C57BL/6J mice expressing Ki-67 was determined by flow cytometry. **f** Individual data points (mice) are shown superimposed upon the mean ± SEM from one independent experiment of two. ****$p < 0.0001$. Statistical significance was determined using the unpaired $t$ test

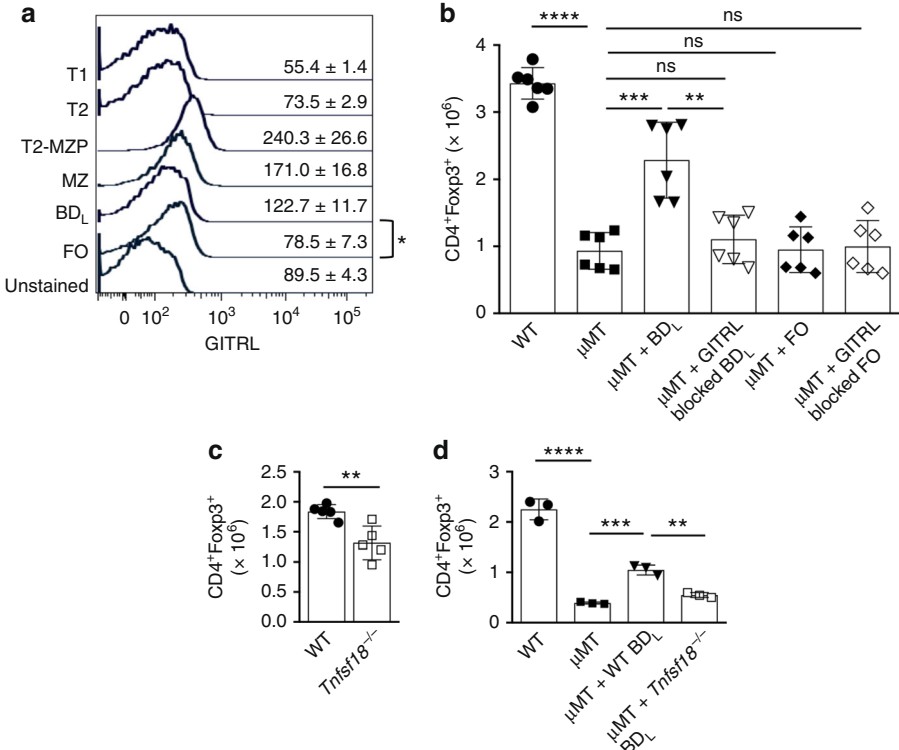

**Fig. 5** $BD_L$ regulatory activity is GITRL-dependent. **a** Splenic T1, T2, T2-MZP, MZ, $BD_L$, and FO B cells from B10.PL mice expressing GITRL was determined by flow cytometry. The data shown are the MFI ± SEM of three mice from one of four independent experiments. *$p < 0.05$, $BD_L$ vs. FO B cells. **b**–**d** C57BL/6μMT mice were reconstituted with $5 \times 10^6$ FACS-purified WT $BD_L$ (**b, d**), WT FO (**b, d**), $Tnfsf18^{-/-}$ (GITRL$^{-/-}$) $BD_L$ (**d**) cells from C57BL/6 mice with (**b**) or without (**d**) GITRL antibody blocking and the absolute number of splenic Treg was determined by flow cytometry 10 days later. **c** Treg were quantitated in the spleen of WT and $Tnfsf18^{-/-}$ mice by flow cytometry. **b** Individual data points (mice) are shown superimposed upon the mean ± SEM from two independent experiments. **$p < 0.01$, μMT + WT $BD_L$ vs. μMT + GITRL blocked WT $BD_L$; ***$p < 0.001$, μMT vs. μMT + WT $BD_L$; ****$p < 0.0001$, WT vs. μMT. **c** Individual data points (mice) are shown superimposed upon the mean ± SEM of five mice. **$p < 0.01$. **d** **$p < 0.01$, μMT + WT $BD_L$ vs. μMT + $Tnfsf18^{-/-}$ $BD_L$; ***$p < 0.001$, μMT vs. μMT + WT $BD_L$; ****$p < 0.0001$, WT vs. μMT. Statistical significance was determined using the unpaired $t$ test

were unable to induce Treg proliferation, as compared to WT $BD_L$ (Fig. 5d). These data indicate that GITRL is the primary mechanism by which $BD_L$ promote Treg proliferation.

**The $BD_L$ subset develops in parallel with FO B cells in the spleen**. We next determined whether $BD_L$ develop independently from FO B cells. This was accomplished by determining the timing of $BD_L$ emergence as compared to FO B cells following sublethal irradiation[18]. As shown in Fig. 6a, b, cells in the T2 stage peaked on day 13 post irradiation. The finding that FO B cells peaked later on day 17 indicates that they developed from the T2 stage (Fig. 6a)[17]. The kinetics of $BD_L$ emergence and peaking was identical to FO B cells, indicating that they develop independently and not directly along the FO pathway (Fig. 6a). These results were confirmed using a bone marrow (BM) transplantation approach (Fig. 6b)[20,21]. In a third confirmatory approach, we adoptively transferred FACS-purified CD45.1$^+$ T2 B cells into CD45.2$^+$ WT mice and determined their maturation into $BD_L$ and FO B cells[18]. One and two days after transfer, T2 cells were detectable in the spleen and they had matured into both $BD_L$ and FO subsets at a similar level (Fig. 6c). The absolute cell number of T2, FO, and $BD_L$ recovered are shown in Fig. 6d, which is consistent with published studies[18].

**$BD_L$ are a stable mature B cell subset**. We next examined the stability of $BD_L$ upon adoptive transfer by repeating the sublethal irradiation experiment (Fig. 6a), and on day 11 just prior to the

emergence of $BD_L$ and FO B cells, FACS-purified CD45.1 $BD_L$ or CD45.1 FO B cells were adoptively transferred into CD45.2 recipient mice. This timepoint was chosen because the splenic microenvironment would support B cell development. The transferred B cells were identified as CD45.1$^+$ and carboxy-fluorescein diacetate, succinimidyl ester (CFSE) was used to determine whether the cells had proliferated. FO B cells upon transfer were stable with 96% of the cells the retaining their phenotype one day later (Fig. 7a) and 88% on day 2 (Fig. 7b). The increase in MZ B cells on day 2 was likely due to shedding of CD23 and not differentiation of FO B cells into the MZ lineage (Fig. 7a, b)[22]. Even though in the steady state only ~5% of FO B cells undergo cell proliferation (Fig. 4f), ~50% of FO B cells underwent proliferation after transfer on day 1 (Fig. 7a, inset), which increased to ~82% on day 2 (Fig. 7b, inset). The induction of FO B cell proliferation was likely due to the lymphopenic environment[23]. When $BD_L$ were adoptively transferred on day 1, ~97% of the cells had undergone proliferation (Fig. 7c, inset) and 86% of the cells exhibited a FO phenotype (Fig. 7c). On day 2, as with the FO B cell subset, the decrease in the number of FO B cells was offset by an increase in MZ B cells (Fig. 7d). Interestingly, the percentage of $BD_L$ remained stable at 12% from day 1 (Fig. 7c) to day 2 (Fig. 7d).

To determine whether the $BD_L$ phenotype is stable long term, we tracked their presence in EAE following adoptive transfer. On EAE day 7, ~5% of splenocytes were B220$^+$ (Fig. 8a). We then gated on IgM$^+$CD23$^+$ cells to exclude CD4$^+$B220$^+$ T cells, isotype class-switched B cells, and plasmablast/plasma cells and

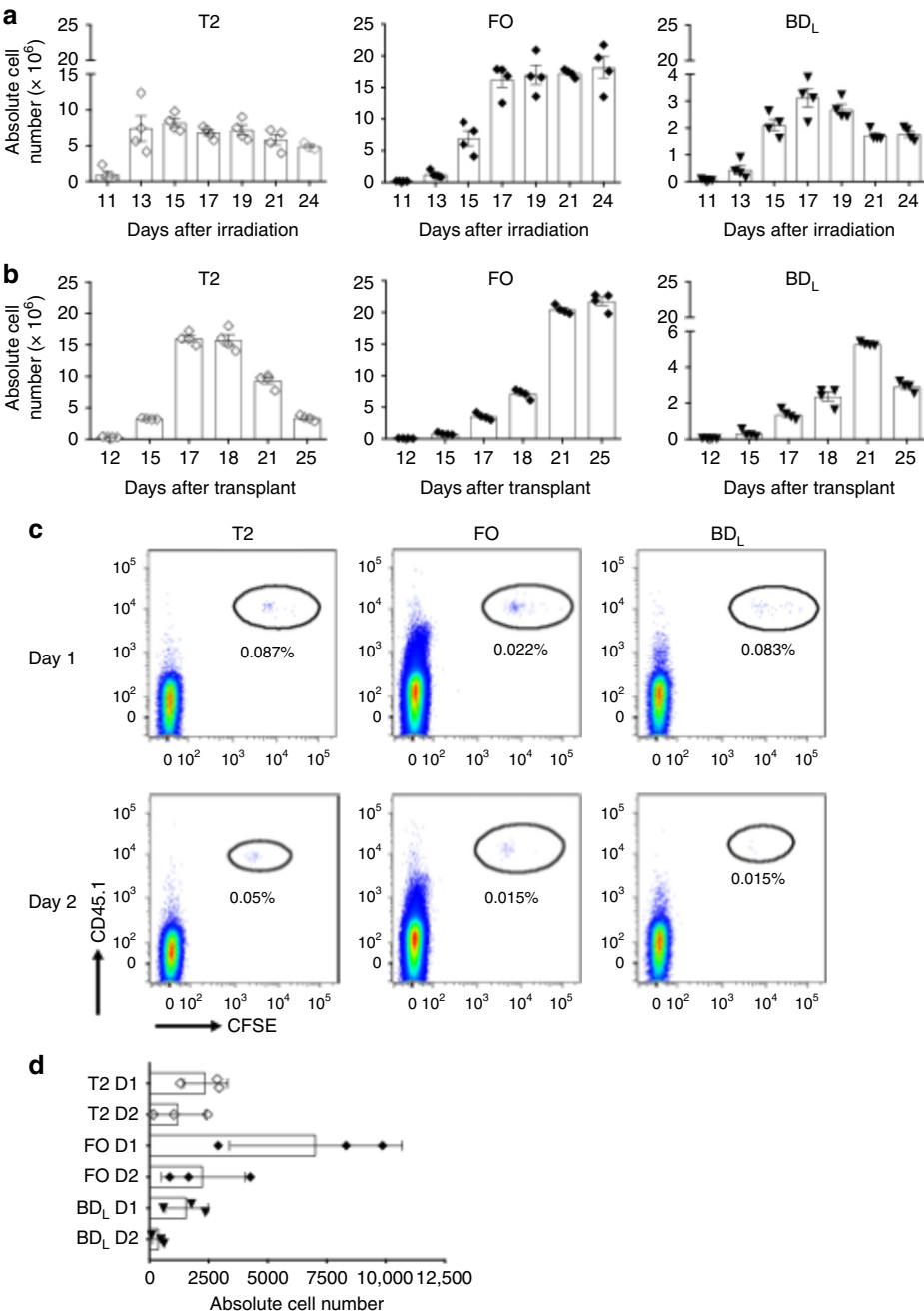

**Fig. 6** $BD_L$ are mature and emerge from the T2 stage. **a** C57BL/6J mice were sublethally irradiated (500 rad) and the emergence of T2, FO, and $BD_L$ was tracked in the spleen on the days indicated by flow cytometry and the absolute number of cells was calculated. **b** C57BL/6J mice were lethally irradiated (950 rad) and transplanted with syngeneic BM and the emergence of T2, FO, and $BD_L$ was tracked in the spleen on the days indicated by flow cytometry and the absolute number of cells was calculated. **a**, **b** Individual data points (mice) are shown superimposed upon the mean ± SEM. **c**, **d** C57BL/6J CD45.1 mice were sublethally irradiated (500 rad) and 15 days later T2 cells were FACS purified, labeled with CFSE, and $1.25 \times 10^6$ cells/recipient were adoptively transferred to C57BL/6J CD45.2 mice and 24 and 48 h later the percentage (**c**) and absolute number (**d**) of T2, FO, and $BD_L$ was determined in the spleen by flow cytometry. T2, FO, and $BD_L$ B cell subsets were gated as for Fig. 2a, b. **c** Data shown are one representative experiment of three. **d** Individual data points (mice) are shown superimposed upon the mean ± SEM from one representative experiment of three. Statistical significance was determined using the unpaired $t$ test

found that ~70% of the cells retained the $BD_L$ $IgD^{low/-}$ phenotype (Fig. 8a). Similar results were obtained at the peak of disease on day 18 (Fig. 8b). On day 30, $BD_L$ were still present (Fig. 8c), which is consistent with Treg levels that were statistically higher in the mice that had received $BD_L$, as compared to μMT mice (Fig. 3d). These cumulative data

indicate that the $IgD^{low/-}$ $BD_L$ phenotype is stable upon adoptive transfer.

**Human $BD_L$ cells are present in spleen and peripheral blood.** We next determined whether a similar population of $BD_L$ human

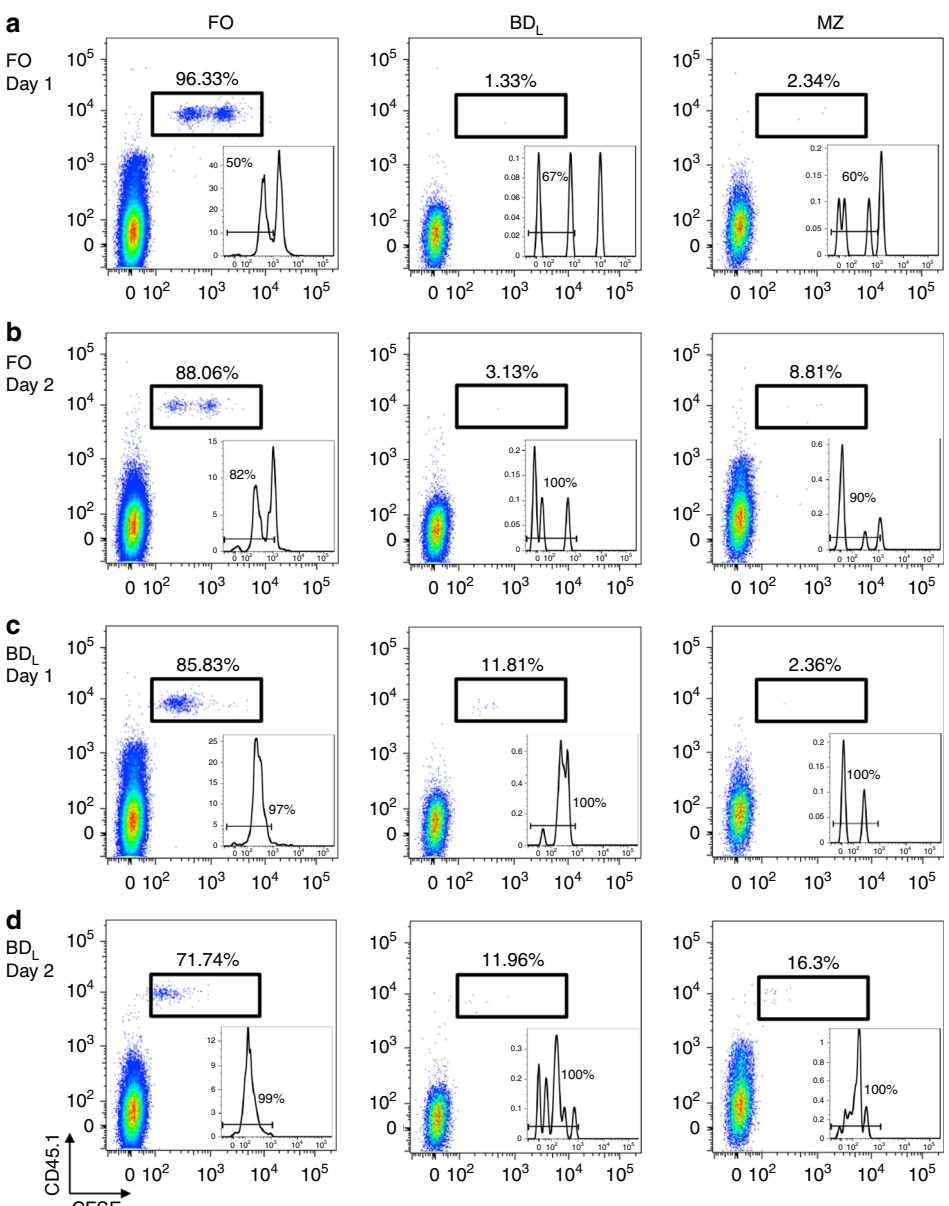

**Fig. 7** Following adoptive transfer a subset of BD$_L$ are stable phenotypically. C57BL/6 CD45.1 FO (**a**, **b**) or BD$_L$ (**c**, d were FACS purified, CFSE labeled, and adoptively transferred (1.5 × 10$^6$) into C57BL/6J CD45.2 mice that were sublethally irradiated (500 rad) 11 days prior. On days 1 and 2 the presence of FO, BD$_L$, and MZ B cells was assessed in the spleen. The transferred cells were gated as CD45.1$^+$ and the percentage of each B cell subset within the gate is provided and shown as dot plots with CFSE. The inset is a histogram of the CFSE data with the percentage of cells having undergone proliferation indicated. Data shown are one representative experiment of four

B cells exist that also promote Treg proliferation. To accomplish this, splenic human B cells were FACS purified by gating on total B cells (CD4$^−$CD19$^+$) (Fig. 9a) and then subsequently on CD20$^+$CD24$^−$ mature B cells (Fig. 9b), from which IgD$^{low/−}$ B cells were obtained (Fig. 9c). We included CD20 in the gating strategy to remove plasma cells. Total B cells and IgD$^{low/−}$ B cells were co-cultured with FACS-purified Tregs (CD4$^+$CD25$^{hi}$) from the same donor in the presence of anti-CD3 and irradiated antigen-presenting cells (APC). We found that Treg proliferation was similar in the presence or absence of total B cells (Fig. 9d). In contrast, in the presence of IgD$^{low/−}$ B cells, Treg proliferation was significantly increased by 50% as compared to Treg alone (Fig. 9d). Figure 9e shows representative CSFE flow cytometry data. We next determined whether we could identify BD$_L$ in the peripheral blood using a modified strategy. By

substituting anti-CD28 for the APC, we largely eliminated Treg background proliferation in the absence of B cells (Fig. 9f). As with the spleen, inclusion of total B cells slightly increased Treg proliferation, and as observed in the mouse, IgD$^{low/−}$ B cells, but not IgD$^{hi}$ B cells supported Treg proliferation (Fig. 9f). Representative gating for Treg and peripheral blood B cells is shown in Suppl. Figure 7. These data demonstrate that human BD$_L$ cells are present in both spleen and peripheral blood that importantly can be identified by an IgD$^{low/−}$ phenotype and induce Treg proliferation.

## Discussion

In this study, we sought to identify a definitive cell surface phenotype that could be used to identify B cells with the capacity to

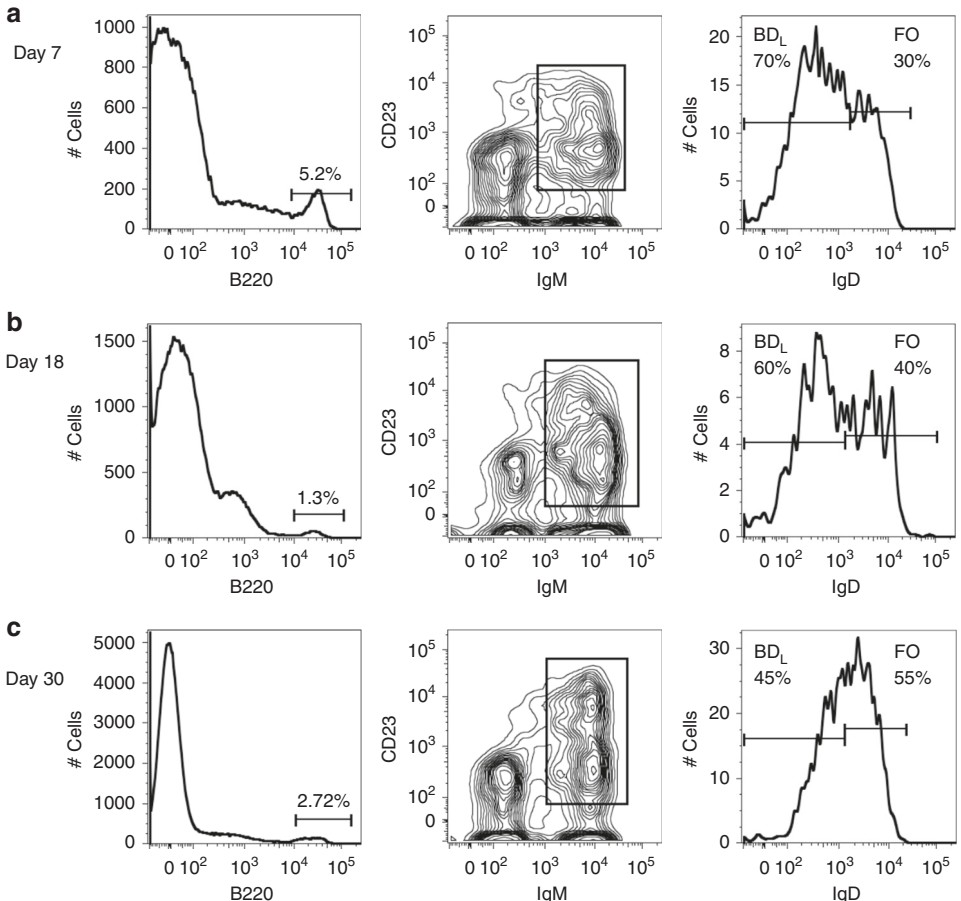

**Fig. 8** Following adoptive transfer and EAE induction BD$_L$ are stable up to 30 days. BD$_L$ adoptive transfer and EAE was performed as for Fig. 3c and the data are from the same mice used for Fig. 3d, e. On days 7 (**a**), 18 (**b**), and 30 (**c**) of EAE, B220$^+$ lymphocytes were gated for IgM$^+$CD23$^+$ and analyzed for IgD expression by flow cytometry. The percentage of BD$_L$ and FO B cells is provided. Data shown are one representative mouse of two

promote Treg proliferation that we have named BD$_L$. Extensive cell surface phenotying and functional assays revealed that BD$_L$ are IgD$^{low/-}$. These cumulative data led to the conclusion that BD$_L$ are a B cell subset that plays an essential role in immune tolerance by regulating the homeostatic expansion of Treg.

The ability of B cells to regulate inflammation was indicated as early as 1974 in models of delayed-type hypersensitivity, sarcoma and autoimmunity[24–26]. These studies fell short of definitively demonstrating a regulatory role for B cells. When μMT mice became available, we demonstrated that B cells were required for EAE recovery[1]. Subsequently, a regulatory role for B cells in a model of inflammatory bowel disease was demonstrated[27], which were shown to regulate via IL-10[28]. B cells were also shown to regulate EAE severity via IL-10[4]. These studies were impactful on the field and led to a plethora of papers demonstrating B cell regulation via IL-10 in a variety of disease models and in humans[29,30]. Given the capacity for all B cell subsets to produce IL-10, in retrospect, it is not surprising that a number of B cell phenotypes were described that regulate via IL-10 production[29–32]. Recently, the phenotype of B cells that regulate via IL-10 was identified as a population of LAG-3$^+$ regulatory plasma cells that develop from numerous B cell subsets in a BCR (B cell Ag receptor)-dependent manner[33]. In humans, the most definitive IL-10-producing B cell phenotype was reported as CD19$^+$CD24$^{hi}$CD38$^{hi}$, which includes both immature and CD5$^+$CD1d$^{hi}$ B cells[34]. Thus, B cells that regulate via IL-10 are not a unique B cell subset.

To define IL-10-independent B regulatory mechanisms, we utilized an adoptive transfer model of EAE, that is, it does not require B cell effector functions[2,3,35]. This approach allowed us to discover that B cells induce the proliferation of Treg in a GITRL-dependent manner[3]. Our development of in vivo B cell regulatory assays allowed us to track the ability of purified B cell subsets to induce Treg proliferation[3]. This afforded us the opportunity to identify a definitive B cell regulatory phenotype. It is well known that the various IgG isotypes exhibit differential cell depletion. The effector function of the IgG subclass is determined by the Fc domain. The primary difference between the mouse IgG$_1$ and IgG$_{2a}$ subclasses is that the later binds with high affinity to both FcγRI and FcγRIV, which are activating receptors that facilitate antibody-dependent cell clearance[36–39]. In mice, anti-CD20 IgG$_{2a}$ efficiently depletes all B cells in the spleen, while the IgG$_1$ depletes the majority of FO B cells, but not the MZ subset (Fig. 1a)[12]. Using the partial depleting strategy, we determined that BD$_L$ activity resided within the undepleted FO subset (Fig. 1g). This result surprised us because at the time B cell regulatory activity had not been described for the FO subset. By performing extensive B cell phenotyping on the non-depleted FO B cells, we found that they were enriched for cells expressing IgD$^{low/-}$ (Fig. 2b). Retained IgM expression (Fig. 2a, d) indicated that they had not undergone isotype class switching leading to loss of IgD[40].

To the best of our knowledge, a subset of FO B cells expressing IgD$^{low/-}$ has not been previously described. Using functional

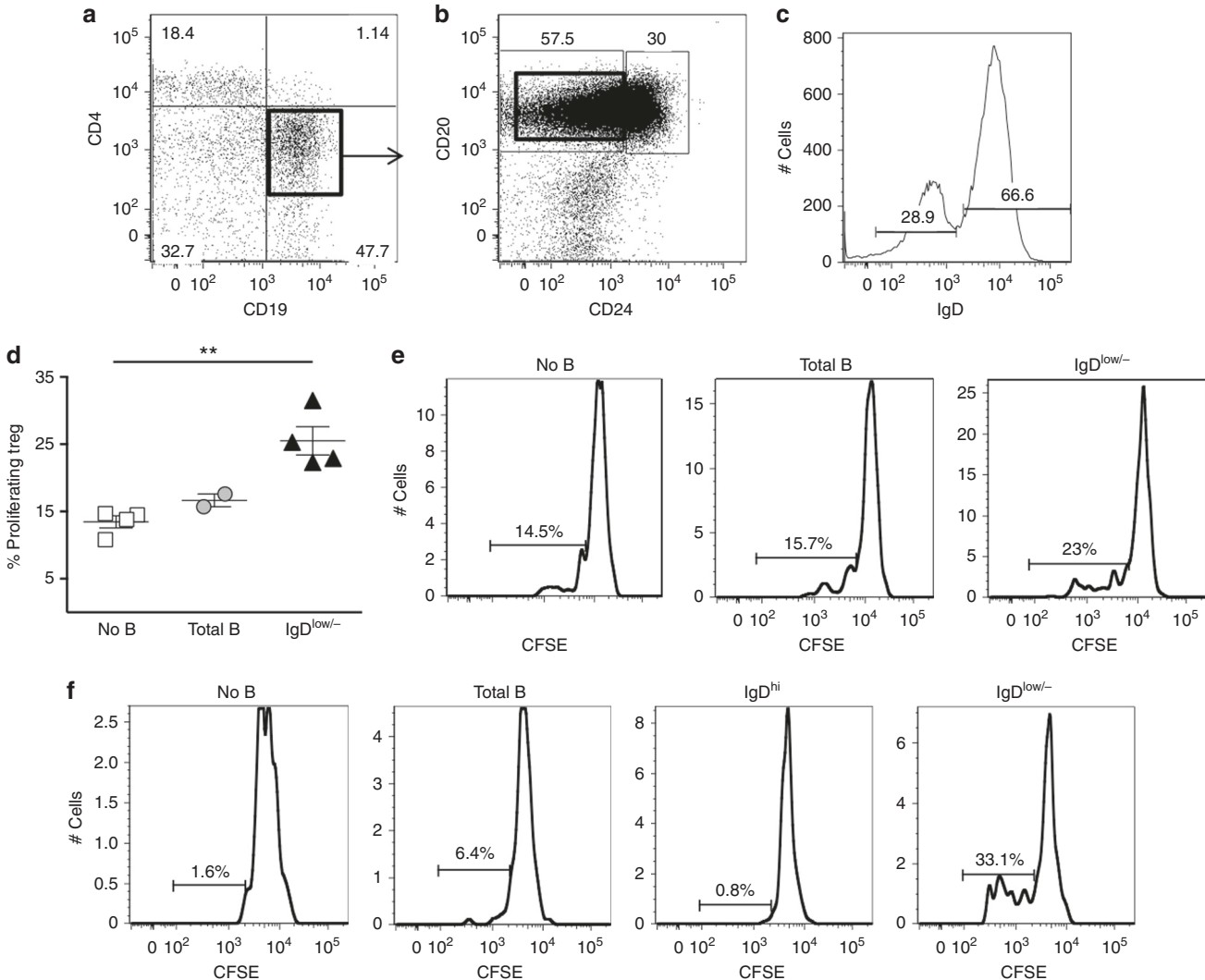

**Fig. 9** Human IgD$^{low/-}$ B cells induce the proliferation of human Treg in vitro. Human splenic B cells (CD19$^+$) (**a**) and IgD$^{low/-}$ (CD19$^+$CD20$^+$CD24$^-$IgD$^{low/-}$) B cells (**a–c**) and Treg (CD4$^+$CD25$^{hi}$) were FACS purified and the Treg were labeled with CFSE. **d** Treg (0.5 × 10$^5$) were cultured alone or with splenic B cells (1 × 10$^5$) in the presence of anti-CD3 (2 mg/ml) and irradiated APC (CD4$^-$CD8$^-$CD19$^-$) for 96 h. Each symbol represents a single human sample. **$p < 0.01$, no B cells vs. IgD$^{low/-}$. **e** Representative flow cytometry histograms with gating of proliferating cells is shown from one experiment of four from **d**. **f** Human peripheral blood B cells (CD19$^+$), IgD$^{hi}$ (CD19$^+$CD20$^+$CD24$^-$IgD$^{hi}$) B cells, IgD$^{low}$ (CD19$^+$CD20$^+$CD24$^-$IgD$^{low/-}$) B cells, and Treg (CD4$^+$CD25$^{hi}$) were FACS purified. CFSE-labeled Treg (0.5 × 10$^5$) were cultured alone or with 1 × 10$^5$ total B cells, IgD$^{hi}$, or IgD$^{low/-}$ in the presence of plate bound anti-CD3 (10 mg/ml) and anti-CD28 (10 mg/ml). After culture, the cells were stained with CD4 and the percentage of proliferating Treg cells was determined by flow cytometry. Data shown are one representative experiment of two. Statistical significance was determined using the unpaired $t$ test

assays, EAE and CHS disease models (Fig. 3), we concluded that BD$_L$ are unique in their ability to induce Treg expansion. As for total B cells[3], BD$_L$ also require GITRL expression to induce Treg expansion (Fig. 5b–d). While GITRL expression is low on all B cell subsets, BD$_L$ expressed significantly higher levels as compared to FO B cells (Fig. 5a). We found that upon adoptive transfer of BD$_L$, ~13–20% of the Treg population underwent cell proliferation. Although Treg numbers increased 2–3-fold (Figs. 3a, 5b), we do not know whether the increase is due to a high proliferation of a small subpopulation of Treg or induction of Treg proliferation on larger scale. However, because the majority of the Treg underwent one to two cell divisions (Suppl. Figures 3B, 4B), we favor the later explanation. We tracked the fate of CD4 T cells during EAE and found both Treg and CD4 T cells were increased in μMT mice that received BD$_L$ (Fig. 3d, e). These data are consistent with a report demonstrating that chronic B cell

depletion during acute viral infection in mice disrupted the homeostasis of both Treg and conventional CD4 T cells[41].

Currently, there are four major populations of mature B cells including B1a that arise from the fetal liver and BM-derived B1b, MZ, and FO subsets[42,43]. CD5$^+$ B1a cells are the major producers of natural IgM and IgG$_3$, aid in the control of infections, play a prominent role in autoimmunity, and produce high levels of IL-10[44,45]. CD5$^-$CD11b$^+$ B1b have the natural tendency to recognize protective antigens in bacteria[46]. MZ B cells are the first line of defense against blood-borne pathogens, particularly those with repeating polysaccharides[47]. FO B cells produce the majority of high-affinity isotype class-switched antibodies in a T cell-dependent manner[48]. Here, we now provide evidence for the existence of BD$_L$ as a fifth mature B2 cell population that develops in the spleen from the T2 stage (Fig. 6). When BD$_L$ were adoptively transferred into an environment that would support their

short-term survival and expansion, we found that 12% of the transferred cells remained stable for 2 days (Fig. 7c, d). The remaining cells upregulated IgD and thus fell into the FO phenotyping gate. When we tracked $BD_L$ following adoptive transfer in EAE just prior to EAE onset, 70% of B cells were $BD_L$ (Fig. 8a). At the peak of disease, this level remained high at 60% (Fig. 8b). As EAE progressed the separation between the $IgD^{low/-}$ and $IgD^{hi}$ B cells became more evident and resembled what was observed in human spleen (Fig. 8, Fig. 9c). On day 30 the separation between $IgD^{low/-}$ and $IgD^{hi}$ B cells became less demarcated more closely resembling naive mice (Fig. 8c). These data suggest that immune responses induce $BD_L$ expansion in parallel with the need for Treg to dampen inflammation.

Adoptive transfer studies demonstrate that a stable $BD_L$ phenotype does exist (Figs. 6–8). However, their relationship to FO B cells is not clear. $BD_L$ could directly differentiate into FO B cells, but we do not favor that hypothesis because the kinetics of the developmental studies show that both subsets emerge simultaneously (Fig. 6a, b). In addition, $BD_L$ may be a subset of FO B cells, but because we have never found a situation in which FO B cells induce Treg proliferation, they are a functionally distinct subset. Figure 10 depicts a revised model of B cell development including $BD_L$. As with other mature B cells, $BD_L$ are endowed with a unique function whereby they play an essential role in tolerance. To that end, we previously showed that total B cell depletion led to a break in tolerance as indicated by reduced numbers of Treg and the onset of spontaneous EAE in MBP-TCR transgenic mice and acceleration of colitis onset in $Il10^{-/-}$ mice[3]. Here, we demonstrate that $BD_L$ induction of Treg expansion is also important for the resolution of CHS (Fig. 3f). Further support for a role for $BD_L$ in regulating disease severity is a recent paper demonstrating that B cells via a GITRL-dependent manner induced Treg expansion that suppressed Friend's virus-specific antibody responses in turn attenuating the magnitude of the germinal center response[49]. These cumulative results provide support for $BD_L$ maintenance of Treg homeostasis being a broadly utilized mechanism in disease.

Because $BD_L$ play an essential role in the maintenance of immunological tolerance they are clinically relevant. In autoimmunity, they could be potentially harnessed to increase Treg numbers[50]. Conversely, a reduction in $BD_L$ thereby decreasing Treg numbers could promote immune responses to clear cancer[51]. Using $IgD^{low/-}$ as the marker of $BD_L$, we have been able to demonstrate the presence of $BD_L$ in humans (Fig. 9). The ability to identify $BD_L$ in the peripheral blood will allow their tracking in disease. The importance of B cells in a variety of diseases has become evident with antibody-based therapeutics to deplete B cells. In MS, B cell depletion with rituximab, an anti-CD20 monoclonal antibody (mAb), has demonstrated efficacy[52,53]. As with mouse, human IgG subclasses also exhibit differential cell-depleting activity and are organized into two groups. Type I lead to CD20 redistribution into lipid rafts triggering complement-dependent cytotoxicity[54,55]. Type II trigger homotypic adhesion and lysosomal nonapoptotic cell death[56–58]. Rituximab is a type I antibody and thus would be predicted to spare some populations of B cells[56]. New-generation type II anti-CD20 antibodies with increased cytotoxicity have been developed, thereby enhancing B cell depletion[59,60]. It is not known whether $BD_L$ are refractory to depletion by anti-CD20 type I and/or II mAb. If $BD_L$ are differentially depleted by anti-CD20 therapeutics, an intriguing concept would be to tailor their use such that Treg numbers are either maintained or reduced depending on the specific disease.

In this study, we provide evidence for the existence of $BD_L$ whose unique effector function is the maintenance of immune tolerance by promoting the homeostatic expansion of Treg in a

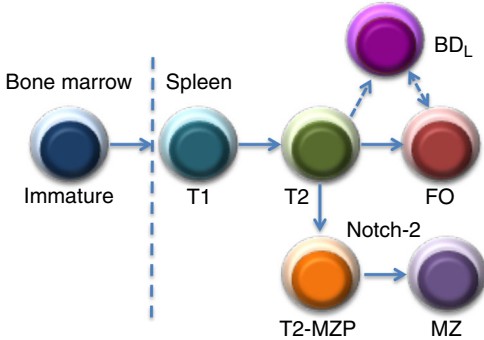

**Fig. 10** Revised model of B cell differentiation in the spleen including the $BD_L$ subset. Currently, it is thought that immature B cells exit the BM and enter the spleen where they undergo sequential differentiation into T1 then T2, which differentiate into either FO B cells or T2-MZP that upon Notch-2 signaling further differentiate into MZ B cells. $BD_L$ are a new mature B cell subset that differentiates directly from the T2 stage. Alternatively, $BD_L$ could be a subset of FO B cells or differentiate from the FO stage

GITRL-dependent manner. Our findings are an important first step in determining the therapeutic potential of $BD_L$.

## Methods

**Mice.** B10.PL, C57BL/6J (CD45.2), B6.129S2-$Ighm^{tm1Cgn}$/J (C57BL/6μMT), B6N (Cg)-$Tnfsf18^{tm1.1(KOMP)Vlcg}$/J (GITRL$^{-/-}$) and B6.SJL-$Ptprc^a$ $Pepc^b$/BoyJ (CD45.1) mice were purchased from The Jackson Laboratory (Bar Harbor, ME, USA). MBP-TCR transgenic mice and generation of Foxp3$^{EGFP}$, μMT, and $Il10^{-/-}$ mice on the B10.PL (H-2$^u$) background was previously described[1,3,61]. MBP-TCR transgenic mice were genotyped by PCR using the following primers: forward-5′-TGC AGT ATC CCG GAG AAG GG-3′; reverse-5′-TTC TCA TTT CCA TAG TTC TCA C-3′. $Notch2^{fl/fl}$CD19cre$^-$ and $Notch2^{fl/fl}$CD19cre$^+$ mice were kindly provided by Dr. Shigeru Chiba (Tokyo University, Japan) and Dr. Maeda Takahiro (Beckman Research Institute of the City of Hope, CA, USA)[16]. Mice were housed and bred at the animal facility of the Medical College of Wisconsin (MCW) and animal protocols using all relevant ethical regulations were approved by the MCW Institutional Animal Care and Use Committee. All experiments were carried out with mice that were age matched and sex matched utilizing both genders.

**B cell depletion in vivo.** Anti-mouse-CD20 mAb (18B12IgG1) and its corresponding isotype control (2B8 msIgG1) were provided by Biogen Inc. Two hundred and fifty micrograms of antibody was intravenously (i.v.) injected into mice, once, or twice, 14 days apart.

**Peptide and antibodies.** MBP Ac$_{1-11}$ peptide (Ac-ASQKRPSQRSK) was generated by the Protein Core Laboratory of the Blood Research Institute, Versiti Wisconsin. The 2.4G2 hybridoma was obtained from American Tissue Culture Collection. Commercial antibodies utilized in this study are described in Supplementary Table 1.

**Flow cytometry and FACS purification.** Single-cell suspensions from mouse and human spleens or human peripheral blood were obtained, counted and 0.2–1 × 10$^6$ cells were cell surface stained with specific antibodies. Frozen human splenocytes were thawed and washed twice prior to staining. Intracellular Foxp3, IgD, and IgM staining was performed using the anti-mouse/rat Foxp3PE staining kit from eBioscience using anti-Foxp3, anti-IgD, or anti-IgM in the final step (San Diego, CA, USA) as per the manufacturer's instructions. Cells were acquired on an LSRII flow cytometer (BD Biosciences, San Diego, CA, USA) and data were analyzed using the FlowJo software (Tree Star Inc., Ashland, OR, USA). Fluorochrome-labeled and/or enhanced green fluorescent protein (EGFP)-expressing cells were sorted using a FACSAria cell sorter (BD Biosciences, San Diego, CA, USA).

**Histolological analysis.** Frozen section (6 μM) of spleens from B10.PL mice treated with 250 μg anti-CD20 antibody (18B12IgG1) or isotype control (2B8msIgG1) were fixed with acetone for 10 min. The fixed splenic sections were stained with anti-mouse IgM-PE-Dazzle 594, IgD-Alex Fluor 488, anti-mouse CD3Brilliant Violet 421, and anti-mouse CD209b (SIGN-R1)-APC and images were captured with Olympus VS120 fluorescent microscope at ×80 magnification.

**EAE induction.** EAE was induced by the i.v. adoptive transfer of 1 × 10$^6$ MBP-specific encephalitogenic T cells activated in vitro with MBP Ac$_{1-11}$ peptide into

sublethally irradiated (360–380 rad) mice as described[2,35]. Clinical symptoms of EAE were scored daily as follows: 0, no disease; 1, limp tail; 1.5, hind limb ataxia; 2, hind limb paresis; 2.5, partial hind limb paralysis; 3, total hind limb paralysis; 4, hind and fore limb paralysis; and 5, death.

**Sorting and adoptive transfer of B cells**. Total splenic B cells from 8–10-week-old mice were purified by negative selection using magnetic cell sorting (STEMCELL Technologies, Vancouver, BC, Canada) followed by FACS purification of FO (B220$^+$IgM$^+$CD21$^{int}$CD23$^+$CD93$^-$) or MZ (B220$^+$IgM$^{hi}$CD21$^{hi}$CD23$^-$CD93$^-$) B cells from anti-CD20-depleted mice or FO (IgD$^{hi}$), MZ, BD$_L$ (B220$^+$IgM$^+$CD21$^{int}$CD23$^+$CD93$^-$IgD$^{low/-}$), or T2 (B220$^+$IgM$^{hi}$CD21$^{low/-}$CD23$^+$CD93$^+$) B cells from WT or $Tnfsf18^{-/-}$ mice. B cell purities were ~99% as determined by flow cytometry. In some experiments, FACS-purified B cells were incubated with anti-mouse GITRL (10 μg/ml) at 4 °C for 60 min for GITRL blocking. B cells were washed in phosphate-buffered saline (PBS) and 5–20 × 10$^6$ cells were i.v. injected into each recipient mouse. EAE or CHS challenge was induced 3 days later or splenic CD4$^+$Foxp3$^+$ cells were enumerated on day 10. For CHS, B cells were transferred one day after sensitization. We have found that BD$_L$ functional activity is equivalent on both the B10.PL and C57BL/6 backgrounds, and thus both strains were utilized in the studies described.

**Induction of CHS**. C57BL/6 mice were sensitized by epicutaneous application of 100 μL 3% (wt/vol) oxazolone (4-ethoxymethylene-2-phenyl-2-oxazolin-5-one) (Sigma-Aldrich, St. Louis, MO, USA) in an acetone:ethanol mixture (1:3 v/v) to shaved abdominal skin. After 4 days, ear thickness was measured using a micrometer. Mice were then challenged by application of 10 μL 1% (wt/vol) oxazolone in an acetone:olive oil mixture (1:1 v/v) to both sides of the ears and ear thickness was measured every 24 h for 5 days.

**RNA-seq and PC analysis**. Total RNA was isolated from sorted B cells using RNAqueous®-4PCR kit (Ambion, Austin, TX, USA). RNA-seq was performed by PerkinElmer (Waltham, MA, USA) using the TruSeq stranded mRNA kit (Illumina, San Diego, CA, USA) on an Illumina HiSeq2500, running HiSeq Control Software V2.2.68, and Real-time Analysis (RTA) software V1.18.66.3. Basecall files (*.bcl) generated by the Illumina instruments were de-multiplexed and converted to fastq.gz format using bcl2Fastq v1.8.2., one file per direction, and two files per sample. For samples that were sequenced on multiple lanes and/or multiple flow cells, resulting fastq.gz files were concatenated into single file sets. The raw fastq files were then mapped to the whole mouse genome build mm9 using Tophat v2.1.0 with parameters --no-novel-juncs --no-coverage-search. Cufflinks were used to estimate the transcript abundance in Fragments Per Kilobase of exon model per Million mapped fragments (FPKM). The average FPKM was calculated for each sample and log$_2$ FPKM was used for PC analysis. Differential expression analysis was conducted with Cuffdiff[62] to calculate the most differentially expressed significant genes based on a Benjamini–Hochberg adjusted $p$ value < 0.05 an absolute fold change of 2 or greater. All significantly differentially expressed genes (log$_2$ fold change <−1 or >1; adjusted $p$ value <0.05) were ordered by descending fold change for gene set overrepresentation analysis. This gene list was submitted to g:Profiler version r1741_e90_eg37[63] as an ordered query to identify overrepresented terms and pathways within the Gene Ontology[64,65] biological process and Kyoto Encyclopedia of Genes and Genomes[66,67] databases.

**B cell development assays**. C57BL/6 mice were sublethally irradiated (500 rad) and on days 11, 13, 15, 17, 19, 21, and 24 the absolute number of splenic T2, FO, and BD$_L$ was determined by flow cytometry. C57BL/6 mice were lethally irradiated (950 rad) and transplanted with 5 × 10$^6$ total BM cells from C57BL/6J CD45.1 mice. C57BL/6J CD45.1 mice were sublethally irradiated (500 rad) and 15 days later splenic T2 B cells were FACS purified and labeled with CFSE (Invitrogen, Carlsbad, CA, USA) prior to adoptive transfer (1.25 × 10$^6$) into C57BL/6J CD45.2 recipient mice. Donor (CD45.1$^+$CFSE$^+$) T2, FO, and BD$_L$ were quantitated 24 and 48 h post transfer. C57BL/6J mice were sublethally irradiated (500 rad), and on day 11 post irradiation, they were adoptively transferred CFSE-labeled FO or BD$_L$ (1.5 × 10$^6$) from C57BL/6J CD45.1 donor mice. Donor (CD45.1$^+$CFSE$^+$) FO and BD$_L$ were quantitated 24 and 48 h post transfer.

**Isolation of human splenocytes and peripheral blood mononuclear cells**. Human splenic tissue was obtained from the Wisconsin Donor Network and Tissue Bank, Versiti Wisconsin (Milwaukee, WI, USA) through written informed consent and approved by the Institutional Review Board (IRB) of Versiti Wisconsin as non-human subject research. Portions of human splenic tissue were digested with collagenase D (1 mg/ml) (Roche) for 1 h followed by dissociation using gentleMACS™ C tubes in combination with the gentleMACS Dissociator (Miltenyi Biotec, San Diego, CA, USA)[68,69]. Red blood cells were lysed with RBC lysis buffer (eBioscience) and lymphocytes were further purified using 60% percoll (Sigma, St. Louis, MO, USA). Cells were washed and stored frozen in liquid nitrogen. Peripheral blood lymphocytes were isolated from buffy coats obtained from Versiti Wisconsin from healthy donors in a non-human subject research manner. Buffy

coats were diluted 2× with PBS and lymphocytes were separated using a 60% Percoll gradient.

**In vitro co-culture of human B cells and Treg**. B cells, APC (CD19$^-$CD3$^-$), and CD4$^+$CD25$^{hi}$ Tregs were FACS purified from human splenocytes or peripheral blood from the same donor. The sorted Tregs were labeled with 3 μM CFSE and cultured alone (0.5 × 10$^5$) or with sorted B cells (1 × 10$^5$) in the presence of anti-CD3 (OKT3) (2 μg/ml) and irradiated (3000 rad) APC or in the presence of plate-bound anti-CD3 (10 mg/ml) and anti-CD28 (10 mg/ml). After 96 h, the cells were stained with CD4, CD25, and DAPI (Thermo Fisher Scientific, Waltham, MA, USA) and proliferation of Tregs was determined by dye dilution and by flow cytometry[3].

**Statistical analysis**. Data were analyzed using GraphPad prism (San Diego, CA, USA) and were presented as mean ± SEM. Statistical significance was determined using the unpaired $t$-test. P values <0.05 were considered significant.

## Data availability

The RNA-seq data have been deposited in the Gene Expression Omnibus under the accession code GSE111911 and went public December 3, 2018. All relevant data and novel reagents are available from the authors upon request.

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

## Acknowledgements

We thank Jennifer Idsvoog for assistance with the animal colony, Benedetta Bonacci for assistance with FACS, Dr. Chiba and Takahiro for the *Notch2*fl/flCD19cre mice, and Biogen Inc. for the anti-CD20 antibody. In particular, we would like to thank the families of the spleen donors whose generosity has allowed our research to advance. This work was supported in part by National Institutes of Health grants 1R56AI122655 (B.N.D.), 1R56AI129348 (B.N.D.), AI069358 (B.N.D.), AI069358 (B.N.D.), AI079087 (D.W.), and HL130724 (D.W.); National Multiple Sclerosis Society research grant RG 1501-03034 (B.N.D.) and the Versiti Blood Research Foundation.

## Author contributions

Conception: A.R., M.I.K. and B.N.D. Experimental design and execution: A.R., M.I.K., S.R., C.J.G. and B.N.D. Data acquisition: A.R., M.I.K., S.B. and D.W. Data analysis and interpretation: A.R., M.I.K., K.L.P., R.T.B., S.R. and B.N.D. Preparation of the manuscript: A.R., M.I.K.., R.T.B., S.R. and B.N.D.

## Additional information

**Competing interests:** The authors declare no competing interests.

