## [Peer Review File · Nature Communications]

Reviewers' comments:

Reviewer #1 (Breg, SLE)(Remarks to the Author):

The authors showed that a novel IgD(low) B cell subset, which increases Treg cell via GITRL. This is well-written paper, however, there are some concerns which have to be answered before publication. My comments are summarized below.

Major.

1. Splenic CD5+ B1 B cells have regulatory capacity and IgD(low) phenotype. Therefore, the author should show CD5 expression of IgD(low) B cell subset, and discuss the difference between IgD(low) B cell and CD5+ B1 B cell.

2. Previous data have reported that IL-6 producing B cell have a disease promoting role in EAE (doi/10.1084/jem.20111675). Thus, the author should investigate IL-6 production from MZ, Fo, IgD(low) B cell.

3. Although the author shows Treg cell proliferation in Figure 5 and 7, the author did not show the effector T cell study. Thus, the author should investigate whether IgD(low) B cells regulate effector T cell proliferation in Figure 5 and 7.

Minor.

In page 11, subtitle, "GTIRL" should be "GITRL".

Reviewer #2 (IL10, Breg)(Remarks to the Author):

Ray et al follow-up on a previous report where they showed that B cell depletion or deficiency in B10.PL mice reduced Tregs, and that both Tregs and recovery from EAE were restored by transfer of B cells in a GITRL-dependent IL-10-independent manner. They now found that partial B cell depletion with an IgG1 rather than IgG2a anti-CD20 does not deplete Tregs in naïve mice nor exacerbate EAE. A protective B cell population remaining after anti-CD20-depletion was found to be a FO B cell population with an IgM^{hi} IgD^{lo} phenotype that reduced EAE and CHS when transferred into B10.PL uMT hosts. These "BDL" cells differentially express many genes vs. conventional IgD^{hi} FO or MZ B cells, including increased mRNA for GITRL and Treg proliferation in a GITRL-dependent manner. A human equivalent was detected in spleen that also enhanced Treg proliferation in vitro. The authors claim that this is a distinct Breg subset, which promotes Tregs and EAE recovery in a GITRL-dependent manner. While the manuscript is very interesting and definitive identification of Breg subsets is long overdue, a number of significant concerns temper enthusiasm.

MAJOR CONCERNS:

1. A number of manuscripts show that Bregs temper EAE (and other autoimmune disorders including contact hypersensitivity) in an IL-10 dependent manner (Fillatreau Nat Immunol 2002; Matsushita JCI 2008; Shen Nature 2014; Matsumoto Immunity 2014; Yanaba Immunity 2008). Others show that B cells deficiency/depletion does not alter Treg number/function and/or that in EAE Bregs and Tregs operate independently (Matsushita JCI 2008 and JI 2010; Hoehlig EJI 2012). These findings raise the question as to whether or not the salient findings here – all performed in B10.PL mice are generalizable to other strains or models – or remain a very interesting but model-specific subset. (To extend this line of reasoning, the disease models here utilize (B10.PL background) uMT mice, known to have abnormal splenic architecture and EAE studies utilize sublethal irradiation and transfer of

transfer activated TCR-Tg T cells into a lymphopenic environment). The human data in and of themselves are not compelling enough (as detailed below) on their own to identify these cells as a distinct Breg subset. This is not at all to say that Bregs do not utilize various mechanisms in different models or to favor one model over others – but it raises questions about the generalizability of these findings, that alters their impact.

2. The gating for IgD^{low} FO B (BDL) cells is clearly arbitrary and this is not a phenotypically distinct subset, despite that suggestion by the authors. The authors also suggest that the RNAseq profile identifies the population as “unique”. This also seems to be an over-interpretation. The PC analysis of BDLs and IgD^{hi} FO B cells is distinct in one direction – but there are no truly distinct genes or transcription factors suggestive of a unique subpopulation. Rather is differential expression (max ~2.5 fold) of genes primarily related to cell cycle and differentiation - suggesting that this population may be an activated phenotype responding to local activation or inflammatory cues.

It is also not at all clear that the BDL population is really a homogeneous subset in terms of function or expression of key functional molecules. For example do they all express similar levels of GITRL, the key functional element identified by the authors. GITRL protein expression must be shown. Is GITRL expressed at uniformly higher levels in BDL than other subsets (especially subpopulations that express similar activation markers). Is the BDL population really a single distinct subset that expresses high GITRL levels, or is this population itself heterogeneous with respect to GITRL levels? In this regard, the population may not be a specific Breg subset - but may simply be enriched for Bregs in the same way that CD9 and TIM-1 enrich for IL-10⁺ Bregs. Of note, both of the latter molecules are at least functionally involved in IL-10 expression and at least in this regard, may better define distinct phenotypic and functional subsets.

3. Most of the functional analysis on BDG cells is performed after anti-CD20 treatment (only Fig 5 appears to be performed on BDL cells freshly isolated from naive mice). Fig 5 only examines Treg levels, which may not completely correlate with Breg function in a disease models (discussed below). B cell depletion may alter GTR expression, or result in repopulation of this subset with newly developed BDL cells from T2 precursors (even conceivably influenced by high BAFF levels typical after B cell depletion). In short, BDL may differ in steady state vs. post-CD20 depletion. Thus, the function of BDL from naive mice should be examined in disease models.

SPECIFIC POINTS

1. Fig 1:

A. While shown by this group 6 years ago, contemporaneous comparison of the current IgG1 anti-CD20 to the previous IgG2a CD20 mAb that causes total depletion of B cells, reduces Tregs, and exacerbates EAE, should be shown to make sure previous findings are still apparent.

B. Fig 1C: The CD20 immunohistology is simplistic and serves little purpose beyond FACs analysis. Immunohistology should aim to demonstrate where B cell depletion really occurs (e.g. mAb doesn't deplete MZ). They should stain IgM vs. IgD and identify the follicles and MZ with appropriate markers.

C. Fig 1E-F: The kinetics of B cell depletion must be shown by the authors in this strain of mice. All they show is that B cells are depleted by 2 weeks in naive mice. Thus, in Fig 1E, we do not know the extent of B cell depletion, relative BDL expansion, or Tregs levels 3 days later when EAE is induced, or at the height of disease on day 14. These mice are irradiated and receive T_H17 cells – all of which could affect B cell depletion differently than in naive mice (d14) in Fig 1 A-D). In their 2012 paper, it appears that Treg depletion was examined at ~14d after EAE induction. Do B cells normalize by d30 when Tregs are shown in 1F?

D. 1E: it appears that some mice got a second dose of anti-CD20. How many mice? Did this affect any parameters? The authors should be more forthcoming.

2. Fig 2:

A. Fig2D: As noted above, IgD expression is continuous so the phenotype is arbitrary. Based on cell cycle, gene expression, and proliferation, this is an activated cell subset. Is it possible that these cells have downregulated/recycled their surface IgD because of BCR engagement? This could be addressed by staining for intracellular IgD.

B. Fig 2E-H: reveals that FO and BDL subsets express each marker as a continuum and no discrete subsets can be identified. Phenotyping shows one representative example. Cumulative data (bar graphs) for MFI should be shown.

C. What are the expression levels of CD20 on the different B cells populations. Does this explain why are MZ and IgD low cells are spared depletion?

3. Fig 3:

A. Do BDL change phenotype or differentiate or after transfer into uMT mice (easily addressed). This is important given claims of a distinct lineage. Also they should at least discuss whether these cells can give rise to plasma cells given two papers in EAE literature showing that PCs are critical for ameliorating EAE?

B. In terms of mechanism of action, it would be critical to show that BDL do not impact disease if Treg are depleted. Is this their only regulatory effect?

C. As mentioned above, do BDL from non-anti-CD20 depleted hosts have same regulatory effect?

D. Fig 3D: In the CHS model Treg levels are shown only at 120h. But disease severity is impacted by BDL by 24hrs. So, do the BDL expand Treg right away? If not, it would suggest that BDL have other mechanisms of action. They should show kinetics of Treg expansion in CHS (and in EAE).

E. In Fig 3: BDL are being added in 10X higher relative number than would be present in transferred total B cells (Fig 3A, Fig 2B) and 5X higher than total FO B cells in 2A ... Do the authors know if less be transferred to see effect?

4. Fig 4:

-As noted above: increased expression of cell cycle genes and high proliferation suggests that these cells are recently activated and may be undergoing differentiation/maturation (see comment 3A). For example, higher levels of Aicda (promotes isotype and somatic hypermutation). Are BDL en route to becoming GC B cells? The authors should comment on expression of markers such as CD44, CD69, CD83, Fas, GL7 (or PNA), CD138, CXCR4, and CXCR5 in their transcriptional analysis.

-Increased proliferation may be stochastic and not a distinct lineage.

-Key differences in expression should be backed up by protein expression level. This is essential for GITR as mentioned above. BDL should be compared to activated FO or MZ B cells and GC B cells.

-Can the authors confirm results of RNAseq were from mice treated with anti-CD20 14 days earlier? The question is whether the transcriptome of BDL in steady state in naïve mice is the same as those found 14 days after CD20-depletion (particularly the proliferation signature).

5. Fig 5

-Knowing GITRL expression on each B cell subset would be key to knowing how much anti-GITRL is being "delivered" to the mice on the transferred B cells.

-Minor: GITRL expressed on other cell types than B cells (5B), but data I 5C is more convincing. The figures show a lot of variability in ability of BDL to expand Bregs vs. WT.

6. Fig 6:

Although both FO and BDL seem to arise at the same time from T2 B cells, this does not prove that there isn't a stochastic process involved in cells expressing lower vs. higher IgD levels. To convincingly demonstrate that these are a distinct subset, the authors would need to KO genes highly expressed in the BDL or FO subset and show that these subsets are differentially spared.

7. Fig 7:

A. The gating of human FO B cells using CD20 vs. CD24 is not a commonly used strategy and as shown, is quite arbitrary. In fact, CD24⁺⁺ TrB cells may be included in the FO gate shown. AS these cells are IgDlo IgM high, they would affect the expression of IgD in the mixed population. For human B cells, CD24 and CD38 are better to more clearly differentiate naive FO B cells.

B. Treg levels:

-In Fig 7D, is the difference in Treg proliferation between BDL and total B cells statistically significant? Why not use IgDhi cells vs. BDL.

-In 7E, B cells are used to present stimulatory Abs. What is FcR expression on IgDhi vs. BDL? Other common costimulatory ligands (e.g. OX40L, CD40, B7)?

8. Discussion:

A. The authors suggest that they chose a model to study Bregs that avoids CFA immunization, thereby limiting the influence of IL-10 induction. This may be true for EAE, but Bregs have been shown to be IL-10 dependent in a number of other models that do not utilize CFA (e.g. transplantation, SLE, CHS, and IBD).

B. Other mechanisms for Breg induction of Tregs have been described including TGFb and IL-10. This should be noted.

Reviewer #3 (Treg/Th cell, T homeostasis)(Remarks to the Author):

This is an interesting paper which examines an important area, namely trying to define further the set(s) of B cells which appear to promote immunoregulation. The authors have done a fine job in their studies, and convincingly shown that a subpopulation of B cells characterized in part by low expression of IgD, exhibit the characteristics of this subset.

1. The paper would benefit from a more consistent use of terminology and definitions. At times the cells are defined as those which promote Treg expansion, yet at other times they are called Bregs, which are sometimes referred to as a "lineage". Personally, I think they are still a long-way from establishing that these are a separate lineage of cells, which would be supported better by exclusive expression of a cell surface marker, or better yet, a differentiation defining transcription factor. This is not to take away from the advance they have made, but I think the claims need to be moderated.

2. Related to #2, can one call a cell a Breg if it doesn't itself exhibit regulatory function independent of another cell type?

3. The delineation of IgDlo vs hi seems rather arbitrary in the mouse FACS plots, in contrast to the human situation where there clearly are two populations. Can the authors comment on this or perhaps show other data where the staining might be a bit different?

4. The authors should comment that the percent dividing differences in the CFSE plots shown in supplemental figure 2a are really quite modest and could be accounted for by just a few proliferating cells.

5. It's essential to show the actual CFSE plots for figures 7D and 7E. It's hard to interpret the differences without seeing an example of the raw data.

6. The paper would be immensely strengthened if the authors could provide data on patients treated with an anti-CD20 mAb showing the effect of this on the IgDlo population.

Rebuttal

We thank the reviewers for the thorough review of our manuscript. In addition, we are pleased that the reviewers found our paper to be well written and the topic of new B cell subset identification of scientific interest. We have responded to all comments. Some of the requested experiments were very insightful and we now propose that the IgD^{low/-} phenotype actually contains two new B cell subsets. We have strong evidence for the existence of BD_L because of function. However, for the second subset, we could only propose that it is a new transitional B cell subset. We appreciate the fact that these new findings bring up many new questions that could be addressed. This is the nature of all new discoveries. However, definitively defining the new subset would take considerable amount of time and is beyond the scope of this manuscript. Thus we have toned down the language in the manuscript to describe BD_L as a new subset, not lineage, and defined them as regulatory in terms of their induction of Treg expansion. Changes to the text are highlighted in yellow.

Reviewer #1

Major

1. Splenic CD5+ B1 B cells have regulatory capacity and IgD(low) phenotype. Therefore, the author should show CD5 expression of IgD(low) B cell subset, and discuss the difference between IgD(low) B cell and CD5+ B1 B cell.

Author response: We have added additional markers of B cells in Fig. 2 (B220, CD20 and CD9) and refined the CD23 and CD21 phenotype (Fig. 2). We also added markers defining IL-10 producing B cells in Suppl. Fig. 2 (CD5, CD1d and Tim 1). These data are referred to in the discussion.

2. Previous data have reported that IL-6 producing B cell have a disease promoting role in EAE (doi/10.1084/jem.20111675). Thus, the author should investigate IL-6 production from MZ, Fo, IgD(low) B cell.

Author response: Differences in IL-6 were not found in our RNAseq data (Fig. 4), but as requested, we performed a qPCR examining IL-6 production and found that both BD_L and FO B cells expressed low levels of IL-6 mRNA with the BD_L levels lower. These data are shown in Suppl. Fig. 4K. From these data we conclude that at least in the steady state, BD_L are unlikely to regulate via IL-6. These data are consistent with B cells not playing a pathogenic role in adoptive transfer EAE.

3. Although the author shows Treg cell proliferation in Figure 5 and 7, the author did not show the effector T cell study. Thus, the author should investigate whether IgD(low) B cells regulate effector T cell proliferation in Figure 5 and 7.

Author response: We have found that conventional T cells also expand in our assay and have now included these data as Fig. 3E along with a kinetic analysis of Treg during EAE (Fig. 3D). These data are consistent with the literature (Lykken, et. al., reference). In the human experiments in Fig. 7 (new Fig. 9) we only put Treg into the assay, thus we cannot determine whether human IgD^{low} B cells will also expand conventional CD4 T cells.

Minor.

In page 11, subtitle, "GTIRL" should be "GITRL".

Author response: This has been corrected.

Reviewer #2:

While the manuscript is very interesting and definitive identification of Breg subsets is long overdue, a number of significant concerns temper enthusiasm.

Author response: We appreciate the very thorough review and have responded to each concern below. We also agree that identification of additional B cell subsets is long overdue and we are very excited to be contributing to that knowledge. Indeed, because of your suggested experiments, we now have evidence of two new B cell subsets.

MAJOR CONCERNS:

1. A number of manuscripts show that Bregs temper EAE (and other autoimmune disorders including contact hypersensitivity) in an IL-10 dependent manner (Fillatreau Nat Immunol 2002; Matsushita JCI 2008; Shen Nature 2014; Matsumoto Immunity 2014; Yanaba Immunity 2008). Others show that B cells deficiency/depletion does not alter Treg number/function and/or that in EAE Bregs and Tregs operate independently (Matsushita JCI 2008 and JI 2010; Hoehlig EJI 2012). These findings raise the question as to whether or not the salient findings here – all performed in B10.PL mice are generalizable to other strains or models – or remain a very interesting but model-specific subset. (To extend this line of reasoning, the disease models here utilize (B10.PL background) μ MT mice, known to have abnormal splenic architecture and EAE studies utilize sublethal irradiation and transfer of transfer activated TCR-Tg T cells into a lymphopenic environment). The human data in and of themselves are not compelling enough (as detailed below) on their own to identify these cells as a distinct Breg subset. This is not at all to say that Bregs do not utilize various mechanisms in different models or to favor one model over others – but it raises questions about the generalizability of these findings, that alters their impact.

Author response: The author makes an interesting point, but all of the studies cited in EAE were performed on the C57BL/6 background. Here, we show that BD_L activity is demonstrable in both C57BL/6 and B10.PL mice. Each figure legend identifies the strain of mouse used. We appreciate the fact that μ MT mice have abnormal splenic architecture and this is a caveat common to all studies performed in this mouse, which is much of the IL-10-regulatory B cell literature. That is why we utilized the anti-CD20 depleting strategy to demonstrate that total B cell depletion leads to a loss of Treg even when the splenic architecture is intact (Ray, et. al., 2012 reference). Because BD_L can be differentiated by both cell surface phenotype and function, we are confident in saying that they are a unique B cell subset. As per reviewer #1, we have dropped the verbiage that they are a unique B cell lineage. BD_L could be a new lineage or a subset of FO B cells, either way they are a newly described subset. Here, we also provide data that BD_L also regulate the extent of contact hypersensitivity (Fig. 3F,G). In addition, a recent paper demonstrated that the expansion in Treg observed after Friend's virus infection required GITRL-expressing B cells (see Moore, et al, 2017 reference). Thus we are confident that our findings are applicable to diseases beyond EAE.

2. The gating for IgD^{low} FO B (BD_L) cells is clearly arbitrary and this is not a phenotypically distinct subset, despite that suggestion by the authors. The authors also suggest that the RNAseq profile identifies the population as “unique”. This also seems to an over-interpretation. The PC analysis of BD_L s and IgD^{hi} FO B cells is distinct in one direction – but there are no truly distinct genes or transcription factors suggestive of a unique subpopulation.

Author response: We disagree that the gating for BD_L is arbitrary. We followed well-established and accepted gating strategies for B cells (Allaman, et. al. 2001 and 2008, Srivastava, et. al. references). B cells are not like T cells in that their subsets do not have distinct markers. Th17 cells are not a stable cell phenotype, which can differentiate into Treg or Th1, and have a very similar cell surface phenotype to Th1 cells, yet they are considered a unique T cell subset. To differentiate the two cell subsets cytokine production and transcription factors are utilized. Without intracellular analysis Th1 and Th17 cells cannot be differentiated by cell surface expression. However, functionally they are different. Here we have a similar situation whereby BD_L and FO B cells are difficult to differentiate based on cell surface expression. However, we have the advantage over Th1 and Th17 cells because we do have a unique cell surface phenotype (IgD^{low/-}) that can be used to purify the two B cell subsets that differentiates them functionally. To isolate live Th1 and Th17 cells for further study reporters have to be utilized. B cells regulate the expression of IgD by several mechanisms. In studies beyond the scope

of this study, we are hopeful that the mechanism utilized by BD_L to downregulate IgD can be utilized as a molecular marker of their identity. There are no known transcription factors that differentiate between B1, FO and MZ B cells yet they are clearly different B cell lineages. CD4 T cells are one of the only cell lineages whose subsets can clearly be defined by transcription factors. Thus the inability to identify BD_L by a transcription factor is not surprising. We went over the PCA with our bioinformatician (coauthor Dr. Robert Burns) and RNA-seq collaborator (coauthor Dr. Sid Rao) to confirm that it could be used as strong evidence that BD_L were unique from FO B cells. Thus we are confident in our conclusion in regards to the PCA. By unique we mean that BD_L can be identified and purified by a combination of cell surface receptors that have not previously been described that define a unique function. In addition, we were the first to demonstrate that B cells promoted Treg homeostasis by inducing their proliferation (Ray, et. al., 2012 reference). Recently, this finding was recently reproduced in Friend's virus infection (Moore, et. al., 2017 reference) providing independent validation of our findings. A discussion of later finding has been added to the Discussion.

Rather is differential expression (max ~2.5 fold) of genes primarily related to cell cycle and differentiation - suggesting that this population may be an activated phenotype responding to local activation or inflammatory cues.

Author response: We do not think that the high level of cell proliferation of BD_L as compared to FO B cells is due to activation for a number of reasons. First, in our developmental studies they emerge from the T2 lineage with FO B cells in the spleen starting on day 13 after irradiation and thus would likely have not been exposed to antigen (Fig. 6). However, we cannot exclude the fact that they are enriched for self-reactivity as for the B1 subset. We are interested in exploring this possibility, but the analysis is beyond the scope of the current manuscript. However, we do know that BD_L are not enriched for either κ or λ BCR as compared to FO B cells (data not shown). Secondly, they do not express higher levels of the activation marker CD25 (Suppl. Fig. 4D). They do express higher levels of CD80 (Suppl. Fig. 4C) and CD86 (Suppl. Fig. 4J) than FO B cells, but this finding alone does not indicate activation. If BD_L were activated, we would have expected to observe increases in cytokine or chemokine production in the RNA-seq data, which we did not observe. In response to reviewer #1 we examined IL-6 a proinflammatory cytokine produced by B cells and it was actually expressed at a lower level than FO B cells (Suppl. Fig. 4K). In addition, we previously demonstrated that neither CD80 or CD86 was required for B cells to induce Treg expansion (Ray, et al, 2012). BD_L are proliferating at a higher rate than both FO and MZ B cells, but equal to the MZP stage (see new Fig. 4F). In response to point 3 under the Specific Points, we performed the experiment requested and now propose that the BD_L phenotyping leads to the purification of the mature BDL subset and a new transitional subset that would likely contribute to the increased proliferation. As discussed in the response to reviewer 3, we are planning to utilize single cell RNA-seq to further identify additional B cell subsets, which may shed light onto the development stage/lineage of BD_L.

It is also not at all clear that the BDL population is really a homogeneous subset in terms of function or expression of key functional molecules. For example do they all express similar levels of GITRL, the key functional element identified by the authors. GITRL protein expression must be shown. Is GITRL expressed at uniformly higher levels in BDL than other subsets (especially subpopulations that express similar activation markers). Is the BDL population really a single distinct subset that expresses high GITRL levels, or is this population itself heterogeneous with respect to GITRL levels? In this regard, the population may not be a specific Breg subset - but may simply be enriched for Bregs in the same way that CD9 and TIM-1 enrich for IL-10+ Bregs. Of note, both of the latter molecules are at least functionally involved in IL-10 expression and at least in this regard, may better define distinct phenotypic and functional subsets.

Author response: GITRL is expressed at very low levels on all B cell subsets. However, we measured GITRL expression by flow cytometry and found that BD_L expressed significantly higher levels of GITRL as compared to FO B cells, albeit the difference was modest (new Fig. 5A). We agree that our BD_L phenotype may enrich for a B cell subset with the intrinsic function of inducing Treg proliferation via GITRL. See comments below regarding the presence of a second unique B cell subset within the IgD^{low/-} population. As discussed in response to Reviewer 3, we plan to assess that possibility using RNA-seq. Even if enrichment has occurred, the function

of the current BD_L phenotype is distinguishable from FO and MZ. Going back to the CD4 T cell analogy above, when Th1 and Th17 are isolated and used in functional assays they are always an enrichment because you never get even close to 100% of the cells producing their signature cytokines. It is now clear that virtually all, if not all, immune cell types are very plastic in that they respond to their microenvironment. This includes monocyte/macrophages, neutrophils, CD4 T cells and so forth. Thus it is conceivable that B cell populations such as FO and MZ will also be composed of subsets. Now that single cell RNA-seq has become more available, we believe that the data will be forthcoming. Thus it remains to be discovered exactly how BD_L fit into the entirety of B cell subsetting but we have proposed a new model of B cell differentiation in Fig. 9. It is an exciting time to do immunology. In regards to IL-10 producing B cells, only recently has a definitive B cell subset been described (see Lino, et. al, reference), yet many papers have been published largely just using their production of IL-10 as the real marker. We are just at the beginning of understanding BD_L.

3. Most of the functional analysis on BDG cells is performed after anti-CD20 treatment (only Fig 5 appears to be performed on BDL cells freshly isolated from naive mice). Fig 5 only examines Treg levels, which may not completely correlate with Breg function in a disease models (discussed below). B cell depletion may alter GITR expression, or result in repopulation of this subset with newly developed BDL cells from T2 precursors (even conceivably influenced by high BAFF levels typical after B cell depletion). In short, BDL may differ in steady state vs. post-CD20 depletion. Thus, the function of BLD from naïve mice should be examined in disease models.

Author response: Additional BD_L phenotyping was performed in naïve mice and are shown in Fig. 2C-I, Suppl Fig. 2C-E and Suppl. Fig. 4A-J. In regards anti-CD20 depleting caveats, that is why the studies in Fig. 3C-D; Fig. 4A-F, Fig. 5A-D and Fig. 8 were all performed using FACS purified BD_L from naïve unmanipulated mice. Thus we are not concerned with the above points regarding anti-CD20 B cell depletion. The use of anti-CD20 was/is indicated in the figure legends where appropriate.

SPECIFIC POINTS

1. Fig 1:

A. While shown by this group 6 years ago, contemporaneous comparison of the current IgG1 anti-CD20 to the previous IgG2a CD20 mAb that causes total depletion of B cells, reduces Tregs, and exacerbates EAE, should be shown to make sure previous findings are still apparent.

Author response: We have performed all of our assays numerous times using both anti-CD20 antibodies and are thus confident in our findings. Because the IACUC 3 Rs regulations include “Reduction”, cumulatively we do not think the above suggested comparison is warranted and would not likely be approved.

B. Fig 1C: The CD20 immunohistology is simplistic and serves little purpose beyond FACs analysis. Immunohistology should aim to demonstrate where B cell depletion really occurs (e.g. mAb doesn't deplete MZ). They should stain IgM vs. IgD and identify the follicles and MZ with appropriate markers.

Author response: We agree and have worked out four-color immunofluorescence to show the loss of IgD expression following anti-CD20 IgG₁ B cell depletion in the spleen. This is shown in the new Fig. 1C. The T cell zone (CD3) and MZ (SIGN-R1) remain intact following partial B cell depletion.

C. Fig 1E-F: The kinetics of B cell depletion must be shown by the authors in this strain of mice. All they show is that B cells are depleted by 2 weeks in naïve mice. Thus, in Fig 1E, we do not know the extent of B cell depletion, relative BD_L expansion, or Tregs levels 3 days later when EAE is induced, or at the height of disease on day 14. These mice are irradiated and receive T_{eff} cells – all of which could affect B cell depletion differently than in naïve mice (d14) in Fig 1 A-D). In their 2012 paper, it appears that Treg depletion was examined at ~14d after EAE induction.

Author response: We did perform a B cell depletion kinetics prior to our identification of the BD_L phenotype. We did not include this previously because we obtained the same result that was already published (Zhang, et. al., 2011 reference). Nevertheless since it was requested, we included it in Suppl. Fig. 1A showing absolute cell numbers for the FO and MZ B cell subsets on days 3, 10 and 14 following administration of anti-CD20 IgG₁. We did not repeat the kinetic study to include BD_L because it is not in compliance with IACUC regulations regarding “Reduction” of mouse usage. Treg numbers are not reduced following B cell depletion, so the comment regarding Treg numbers on day 3 when EAE is induced is not applicable. In our 2012 paper, we also performed a kinetic B cell depletion, which was consistent with the literature. For the reviewer’s information the anti-CD20 IgG_{2a} antibody results in depletion of ~90% of B cells by day 3, which includes both FO and MZ B cells. By day 10 virtually all B cells in the spleen are depleted.

Do B cells normalize by d30 when Tregs are shown in 1F?

Author response: We have removed the old Fig. 1F and replaced it with a new kinetic study showing Treg (Fig. 3D), conventional CD4 T cells (Fig. 3E) and BD_L (Fig. 8) during the EAE time course.

D. 1E: it appears that some mice got a second dose of anti-CD20. How many mice? Did this affect any parameters? The authors should be more forthcoming.

Author response: In Fig. 1E, we clearly show with arrows when the anti-CD20 treatment was given. This information is also provided in the Materials and Methods and in the figure legends. Thus we are unclear why the reviewer states that we were not forthcoming. Our anti-CD20 depletion strategy was based on previously published studies (Zhang, et. al., 2011 reference).

2. Fig 2:

A. Fig2D: As noted above, IgD expression is continuous so the phenotype is arbitrary.

Based on cell cycle, gene expression, and proliferation, this is an activated cell subset. Is it possible that these cells have downregulated/recycled their surface IgD because of BCR engagement? This could be addressed by staining for intracellular IgD.

Author response: See response above regarding the arbitrary comment and activation. It is possible that IgD was downregulated due to BCR engagement, which could be due to self-reactivity. We are also curious as to why BD_L express low levels of IgD, which was not due to lack of intracellular IgD (Fig. 2C). However, as compared to FO B cells, BD_L expressed lower levels of intracellular and cell surface IgD (Fig. 2C), indicating that there is an active process limiting IgD expression in BD_L. At this time we cannot explain why surface IgD levels are lower, but we are actively pursuing the question. We also stained for intracellular IgM, which was similar to FO B cells (Fig. 2D). During B cell development the transitional stages are thought to engage self-antigens for selection purposes and undergo expansion to increase cell numbers. Thus type of proliferation is not considered activation. A parallel occurs in T cell development whereby expansion in the thymus after selection is not activation. Thus we do not agree that the proliferation is an indicator of activation in our studies.

B. Fig 2E-H: reveals that FO and BDL subsets express each marker as a continuum and no discrete subsets can be identified. Phenotyping shows one representative example. Cumulative data (bar graphs) for MFI should be shown.

Author response: We disagree that bar graphs are the best method in which to display flow cytometry data, which is why we always show representative staining. We have now included the mean +/- SEM for Fig. 2C-I on each, which demonstrates a low level of variability among nine mice. B cell phenotyping is largely a continuum, a concept that has been established and is well accepted in the field (see Allman and Pillai, 2008; Hardy and Hayakawa, 2001, Allman et al., 2001; Srivastava et al., 2005 references). Thus we are not concerned with our phenotyping strategy. Unfortunately, unlike T cell lineages, definitive markers such as CD4, CD8 and NK1.1 are not available. That makes B cells studies much more challenging and why additional B cell subsets

are so difficult to identify.

C. What are the expression levels of CD20 on the different B cells populations. Does this explain why are MZ and IgD low cells are spared depletion?

Author response: We have included this data in Fig. 2F. CD20 levels do not vary much on B cell subsets throughout development and into activation. Thus we were not surprised to find that BD_L express CD20 similarly to FO and MZ B cells. Antibody depletion of cells is largely dependent upon the effector function of various antibody isotypes not cell surface expression levels. This concept is outlined in the discussion.

3. Fig 3:

A. Do BDL change phenotype or differentiate or after transfer into uMT mice (easily addressed). This is important given claims of a distinct lineage. Also they should at least discuss whether these cells can give rise to plasma cells given two papers in EAE literature showing that PCs are critical for ameliorating EAE?

Author response: We disagree that assessing a change in phenotype is easily addressed. You cannot adoptively transfer a small population of B cells into WT mice because they will not engraft well and thus are difficult to track. In addition, in the absence of inflammation adoptively transferred B cells undergo a short burst of homeostatic expansion in lymphopenic recipients and then die. B cell do not continuously homeostatically expand like T cells. Thus we needed to develop a strategy based on the literature. To address the stability question, we FACS purified FO B cells and BD_L and adoptively transferred them into WT mice 11 days after sublethal irradiation. This strategy was based on Fig. 6A. WT mice were used to preserve the splenic architecture and the irradiation was to make space for the incoming B cells. We waited until day 11 after irradiation because we knew the splenic environment was supportive of FO B cell and BD_L development/survival. These data are shown in Fig. 7. As expected the FO phenotype was relatively stable over two days (Fig. 7A,B). After BD_L transfer 12% of the cells on both day 1 and 2 retained low/neg levels of IgD. The remaining cells had upregulated IgD. These data are difficult to interpret for a variety of reasons. Given that BD_L have intracellular IgD whatever mechanism prevents its cell surface expression could be altered upon the cells entering a new environment, i.e. they may not migrate to the correct niche within the spleen. We have not determined whether low/neg levels of IgD are required for the BD_L phenotype. Thus even though they upregulate IgD they may still induce Treg proliferation. Lastly, we do not know if our BD_L phenotype represents one cluster of cells or multiple. This was discussed above and is our favored hypothesis, which has been included in Discussion and incorporated into a new model of B cell differentiation in the spleen (Fig. 9). Finally, we tracked BD_L during EAE after adoptive transfer and found that they were a stable phenotype until day 30 (Fig. 8).

In regards to plasma cells, this cannot be assessed in our EAE model because antibody responses are not major component of the model. That assessment would require a different disease model and is beyond the scope of the current study. We do have the experiment planned and will address the PC question in the Friend's virus model (see Moore, et al, 2017 reference). In addition, in the studies in which PC were identified as regulatory they did so by IL-10. BD_L do not regulate via IL-10.

B. In terms of mechanism of action, it would be critical to show that BDL do not impact disease if Treg are depleted. Is this their only regulatory effect?

Author response: That experiment cannot be done because Treg depleted mice will very quickly succumb to autoimmunity (see Rudensky, 2011 reference).

C. As mentioned above, do BDL from non-anti-CD20 depleted hosts have same regulatory effect?

Author response: As discussed above, they do.

D. Fig 3D: In the CHS model Treg levels are shown only at 120h. But disease severity is impacted by BDL by 24hrs. So, do the BDL expand Treg right away? If not, it would suggest that BDL have other mechanisms of action. They should show kinetics of Treg expansion in CHS (and in EAE).

Author response: We have performed an EAE kinetic study examining Treg and CD4 effector cell numbers throughout the time course. These data are shown in Fig. 3D,E. Because the CHS experiment was just to demonstrate that BDL activity is not unique to EAE, we did not perform the kinetics of Treg expansion in that model. We have not ruled out that BDL have other mechanisms, which would not surprise us since no immune cell populations seems to have only one function. However, the study of alternative functions is beyond the scope of the current study.

E. In Fig 3: BDL are being added in 10X higher relative number than would be present in transferred total B cells (Fig 3A, Fig 2B) and 5X higher than total FO B cells in 2A ... Do the authors know if less be transferred to see effect?

Author response: We performed a dose-response curve and found that similar Treg expansion was observed after transfer of 2 or 3 x 10⁶ BDL (Suppl. Fig. 2), but was maximal with 5 x 10⁶ BDL (see Fig. 3A). This result was expected for two reasons. First, because ~2 x 10⁶ BDL are present in the spleen and only a fraction of the transferred B cells would localize to the spleen after i.v. transfer. In addition, we now believe that the BDL phenotyping strategy leads to the purification of BDL as well as a transitional B cell subset. If this is the case then even lower numbers than 2-5 x 10⁶ cells are required to measure BDL regulatory activity.

4. Fig 4:

-As noted above: increased expression of cell cycle genes and high proliferation suggests that these cells are recently activated and may be undergoing differentiation/maturation (see comment 3A). For example, higher levels of Aicda (promotes isotype and somatic hypermutation). Are BDL en route to becoming GC B cells? The authors should comment on expression of markers such as CD44, CD69, CD83, Fas, GL7 (or PNA), CD138, CXCR4, and CXCR5 in their transcriptional analysis.

Author response: As discussed above BDL are likely not activated and on their way to becoming PC. In the steady state there is no antigen that would be driving PC differentiation and antibody responses are not required for Treg homeostasis or EAE pathogenesis in our studies. Thus we did not phenotype for plasma cell markers. Furthermore, even if plasma cells were present, which they are likely not, we could not use the markers utilized to differentiate BDL, FO and MZ B cells to track the origin of the plasma cells to any particular B cell subset. CXCR4 and CXCR5 transcriptional levels will not provide information regarding their function in terms of migration. There is crosstalk among chemokine receptors and there are both positive and negative regulators of their function. Thus qPCR data for chemokine receptors is not very useful to determine their function. We are examining the function of various chemokine receptors on BDL, which we plan to include in a separate paper and have thus not included them in this study. However, we have included a flow cytometry analysis of a number of additional cell surface markers (Fig. 2C-I, Fig. 5A, Suppl. Fig. 2C-E and Suppl. Fig. 4).

-Increased proliferation may be stochastic and not a distinct lineage.

Author response: We agree with this comment and did not include language in the text that indicated we utilized proliferation to conclude lineage. We simply showed what the RNA-seq data revealed. We have expanded the proliferation data to include transitional as well as mature B cell subsets in the spleen in Fig. 4F. As per reviewer 1, we have tempered our conclusion regarding lineage and have thus solely utilized the term subset.

-Key differences in expression should be backed up by protein expression level. This is essential for GITR as mentioned above. BDL should be compared to activated FO or MZ B cells and GC B cells.

Author response: We assume the reviewer means GITRL, which was discussed above and shown in Fig. 5A. We do not know why BDL should be compared to GC B cells, since GC B cells will have downregulated most of the cell surface proteins required to phenotype BDL. The partially activated phenotype of MZ B cells in the steady state is well known as demonstrated by their high expression of CD80 (Suppl. Fig. 4C), CD86 (Suppl. Fig. 4J), CD38 (Suppl. Fig. 4H) and CD40 (Suppl. Fig. 4I). Please note that BDL expression of CD80, CD86, CD38 and CD40 are more similar to FO than MZ B cells, further providing support that they are not activated. Finally, the phenotype of activated FO B cells has been very well described in the literature and depends on the nature of the activation signal. Thus we feel that such a comparison is not necessary. As for other proteins, we have now included a number of additional cell surface expression profiles comparing BDL to FO and MZ B cells (Fig. 2C-I, Fig. 5A, Suppl. Fig. 2C-E and Suppl. Fig. 4). The data demonstrate that BDL are more similar to FO B cells than MZ.

-Can the authors confirm results of RNAseq were from mice treated with anti-CD20 14 days earlier? The question is whether the transcriptome of BDL in steady state in naïve mice is the same as those found 14 days after CD20-depletion (particularly the proliferation signature).

Author response: The RNAseq was performed using WT unmanipulated mice. Whenever anti-CD20 depletion was utilized it is stated in the figure legend.

5. Fig 5

-Knowing GITRL expression on each B cell subset would be key to knowing how much anti-GITRL is being “delivered” to the mice on the transferred B cells.

Author response: This has been addressed above and the flow data are shown in Fig. 5A. TNF-family receptors, such as GITRL, are typically expressed at very low levels and thus protein levels are difficult to measure. Thus functional assays such as in Fig. 5B-D are required.

-Minor: GITRL expressed on other cell types than B cells (5B), but data I 5C is more convincing. The figures show a lot of variability in ability of BDL to expand Bregs vs. WT.

Author response: We assume the reviewer means Treg and not Breg. Yes, there is some variability in the level of expansion, but this is the nature of in vivo immunological experiments in which variables that have to do with live organisms cannot be controlled. However, BDL always induce Treg proliferation, whereas FO and MZ B cells do not. In addition, within each individual experiment the SEM are very small indicating consistency with the assay.

6. Fig 6:

Although both FO and BDL seem to arise at the same time from T2 B cells, this does not prove that there isn't a stochastic process involved in cells expressing lower vs. higher IgD levels. To convincingly demonstrate that these are a distinct subset, the authors would need to KO genes highly expressed in the BDL or FO subset and show that these subsets are differentially spared.

Author response: The experiment you just described is the holy grail for the B cell field. While MZ B cell development can be prevented by knocking out Notch-2, no such gene has been found for FO B cells. Thus unfortunately at this time the experiment suggested cannot be done. We agree that IgD levels may be a stochastic process and we are currently investigating that possibility. While IgD^{low/-} is a marker of BDL, we have yet to determine whether it also reflects function. We do not believe that it does but the studies are too preliminary to comment on and are beyond the scope of this manuscript.

7. Fig 7:

A. The gating of human FO B cells using CD20 vs. CD24 is not a commonly used strategy and as shown, is quite arbitrary. In fact, CD24⁺⁺ TrB cells may be included in the FO gate shown. AS these cells are IgD^{lo} IgM

high, they would affect the expression of IgD in the mixed population. For human B cells, CD24 and CD38 are better to more clearly differentiate naïve FO B cells.

Author response: In the human studies, no specific markers were utilized to identify FO B cells thus we are unclear what is meant by the above comments. We disagree that using CD24 to identify human B cells is arbitrary. It was used to identify all mature B cells, which are CD24⁺. CD20 was used to gate out PC, which are CD20⁻. Thus CD24⁺CD20⁺ B cells are all of the possible mature B cell subsets. Based on our mouse phenotyping this is the population in which we would expect to find BD_L regulatory activity, which we did. We were not interested in identifying FO B cells because in the mouse this is not the population in which BD_L regulatory activity resides. CD38 expression on mature B cells can be +/- depending on the B cell stage from naïve to post-GC, thus it is not a reliable marker for mature naïve B cells (see doi: [10.1371/journal.pone.0162209](https://doi.org/10.1371/journal.pone.0162209)).

B. Treg levels:

-In Fig 7D, is the difference in Treg proliferation between BDL and total B cells statistically significant? Why not use IgD^{hi} cells vs. BDL.

Author response: We did not perform the statistics on the total B versus IgD^{low/-} because the total B cell group only contains two data points. As for why we did not use IgD^{hi} in that experiment is largely due to logistical timing of the experiments. It took us some time to work out the conditions for the human in vitro Treg proliferation assay. We tried very hard to find a definitive BD_L phenotype in humans using a variety of markers including CD38 (see above) but found that BD_L activity resided within many phenotyping strategies. In addition, we did not have access to many human spleens so we needed a source of additional cells. That is why we changed to peripheral blood. In addition, peripheral blood had the advantage of it being the source of B cells we would use to track BD_L during disease. Thus when we switched to peripheral blood we took a step back and decided to just use the one defining marker we found in mouse (IgD^{low/-}) to see if the same marker can identify human BD_L activity, which it did. In our discussion of the data in Fig. 7 (new Fig. 9), we did not state that we found a definitive human BD_L phenotype, rather that we can detect their activity.

-In 7E, B cells are used to present stimulatory Abs. What is FcR expression on IgD^{hi} vs. BDL? Other common costimulatory ligands (e.g. OX40L, CD40, B7)?

Author response: B cells only express FcγRII, which is not a stimulatory FcR, but inhibitory. FcγRI and FcγRIII are the stimulatory receptors, which are expressed by myeloid cells. In 7E, the antibodies were coated on the surface of the microtiter plate. We have not stained for OX40L, but CD40, CD80 and CD86 are shown in Suppl. Fig.4, along with additional markers.

8. Discussion:

A. The authors suggest that they chose a model to study Bregs that avoids CFA immunization, thereby limiting the influence of IL-10 induction. This may be true for EAE, but Bregs have been shown to be IL-10 dependent in a number of other models that do not utilize CFA (e.g. transplantation, SLE, CHS, and IBD).

Author response: The above statement is true. However, in SLE and IBD there is a strong role for TLR signaling, which strongly induces IL-10 production by B cells. Nevertheless, we chose a strategy whereby B cell regulation by IL-10 was absent or dampened. Because we needed to expand the discussion, we largely removed much of the text that referred to CFA.

B. Other mechanisms for Breg induction of Tregs have been described including TGFβ and IL-10. This should be noted.

Author response: We assume the authors mean the generation of inducible Treg (iTreg). That is not what we are studying. What we are measuring is induction of Treg proliferation, not induction of Foxp3 thereby generating iTreg. We concluded that in our previous publication (see Ray, et al., 2012 reference).

Reviewer #3

1. The paper would benefit from a more consistent use of terminology and definitions. At times the cells are defined as those which promote Treg expansion, yet at other times they are called Bregs, which are sometimes referred to as a “lineage”. Personally, I think they are still a long-way from establishing that these are a separate lineage of cells, which would be supported better by exclusive expression of a cell surface marker, or better yet, a differentiation defining transcription factor. This is not to take away from the advance they have made, but I think the claims need to be moderated.

Author response: We have gone through the manuscript and tightened up the language. We agree that we have not definitively established that BD_L are a new lineage, so we changed the language to subset. Given that only BD_L induce the homeostatic expansion of Treg and they exhibit a unique cell surface phenotype, we feel that subset is appropriate. We are working on determining whether we can define BD_L using transcription factors, but this is difficult because there are no transcription factors that can differentiate between B1, FO and MZ B cells. Thus the best we can do is determine whether particular transcription factors that are more highly expressed in BD_L as compared to FO B cells control their regulatory activity. At this point in our studies, we have opted not to show specific transcription factor validation data because they may not turn out to be important in the phenotype or function of BD_L . There are some transcription factors listed in the graphs in Fig. 4C-E, so the readers of the manuscript will have some information on their potential transcriptional regulation. In addition, the RNA-seq data has been loaded to the Gene Expression Omnibus and will go public March 1, 2019, as indicated in the acknowledgments.

2. Related to #2, can one call a cell a Breg if it doesn't itself exhibit regulatory function independent of another cell type?

Author response: The reviewer makes an interesting point. We agree that the Breg terminology has been used quite loosely in the B cell field. We did not coin the Breg terminology and have opted to remove it from the entire manuscript upon reflection.

3. The delineation of IgD^{lo} vs IgD^{hi} seems rather arbitrary in the mouse FACS plots, in contrast to the human situation where there clearly are two populations. Can the authors comment on this or perhaps show other data where the staining might be a bit different?

Author response: In regards to the human the IgD^{low} population will include B cells that have undergone isotype class switching. These likely outnumber the BD_L contribution to the population and thus you get a clear distinction. Our mouse studies were all done on young naïve mice that would have undergone minimal isotype class switching. We agree that our cut off for IgD seems arbitrary, but we kept that gating because it functionally defined the ability to induce Treg expansion or not. In addition, if we cut the gate more conservatively it would have been difficult to obtain enough cells for our experiments by FACS. **[Redacted]** Interestingly, after adoptive transfer during EAE the $IgD^{low/-}$ versus IgD^{hi} become more distinct and more closely resembled the human situation (compare Fig. 8 with Fig. 9). We have included these points in the Results and Discussion.

4. The authors should comment that the percent dividing differences in the CFSE plots shown in supplemental figure 2a are really quite modest and could be accounted for by just a few proliferating cells.

Author response: We agree with this comment and have added language to the manuscript in the discussion to clarify the data.

5. It's essential to show the actual CFSE plots for figures 7D and 7E. It's hard to interpret the differences without seeing an example of the raw data.

Author response: We agree and have added the raw flow cytometry data to the figure, which is now Fig. 9.

6. The paper would be immensely strengthened if the authors could provide data on patients treated with an anti-CD20 mAb showing the effect of this on the IgDlo population.

Author response: We absolutely agree with this comment and are planning to do the suggested study. We have set up the collaboration, but have not done the studies because we still lack a definitive BD_L cell surface phenotype in humans. We described our strategy to achieve that goal in point 3, which will take a substantial effort to validate. Thus in writing this manuscript we opted to include the finding that BD_L activity exists in humans. One always has to balance getting the story out with impact.

Reviewers' comments:

Reviewer #1 (Remarks to the Author):

The authors have revised the manuscript appropriately.

Reviewer #2 (Remarks to the Author):

Ray et al now identify a subpopulation of FO-like B cells that express low levels of IgD (BDL) that appear to be the B cells responsible for their previously reported findings. These "BDL" cells differentially express many genes vs. conventional IgD^{hi} FO or MZ B cells, express increased GITRL, promote Treg proliferation and inhibit EAE and CHS in uMT mice. A human equivalent was detected that also enhance Treg proliferation in vitro. Additional explanation and figures along with dropping the claim that this is a distinct lineage have significantly improved the manuscript. However, concerns remain over the mechanism and generalizability that dampen enthusiasm for publication in Nature Communications.

Major Concerns:

1. BDL are not distinct subset: While IgD^{lo} FO B cells are a subpopulation based on this phenotype and their function, they are not a "distinct subset" as claimed by the authors. The expression of IgD is a continuum and there is no distinct phenotypic subset as claimed on pg. 9. In their rebuttal, the authors note that other B cell subsets are not cleanly separated phenotypically. Nonetheless, markers used in combination allow most of the widely accepted B cell subsets to be separated and moreover, these also differ in their development, localization and chemotaxis within the SLO, activation/survival cues, and in some cases (plasma cells, GC B cells) transcription factors and signaling pathways (Notch2:MZ). Rather, BDL cells appear to be a subset of FO B cells that are at the end of IgD expression continuum that exhibit regulatory function. The cut-off between these cells and standard FO cells is arbitrary and with no reason to draw the gate in one place versus another (hence not distinct). If one examined the transcription profile of B cells just on either side of the cursor in Fig 2B, it is unlikely that they would give a dramatic step-off seen when the entire populations of FOB and BDL cells is assessed. The differences in transcriptional profile are primarily in genes related to proliferation and cell cycling and thus it cannot be claimed this population is a distinct subset (See below). This agrees with the noted increase in BDL homeostatic proliferation. New phenotyping in Fig S4 are not able to further differentiate these subsets because differences with FO B cells are marginal. Finally, the new data in Fig 7 and 8 suggest that some BDL can convert into and may be in equilibrium with FOB cells – again suggesting that BDL are not a distinct subset. Taken together these do not support – again not a distinct subset. By comparison, Kahn et al (Nat Com 2015) are extremely careful not to call their PD-L1 high B cells that regulate Tfh and humoral immunity a distinct subset. Bao et al (Cell Research, 2014) describe a phenotypically distinct population of CD11a/FCgRII^{hi} B cells with distinct transcription factor expression and are still reserved – calling it a new subset. The words distinct cannot be ascribed to this subset and data used to push for this designation are generally over-interpreted.

2. Interpretation of transcriptome data: The differences in transcriptome identified by RNAseq as noted above, are in a single dimension and are primarily related to cell cycling and proliferation suggesting that this is a subset of FO cells that is activated and/or undergoing higher homeostatic proliferation. This reviewer does not agree with the author's rebuttal that there is no B cell activation in naïve mice – they have GCs and have Plasma cells due to responses to environmental and endogenous antigens. REGARDLESS, B cells (especially BDL and transitional B cells) do undergo homeostatic proliferation in naïve mice and especially in lymphopenic hosts – as shown by the authors.

Thus, BDL appears to be a subset of FO B cells undergoing proliferation and is possibly even auto-reactive.

3. Mechanism is not proven: The regulatory activity of BDL is correlated with an increase in Tregs – but this mechanism is never proven. Moreover, BDL may have regulatory effects not related to Tregs. This could be examined by depleting or blocking Tregs, for example with anti-CD25 (no breeding of new strains required). The authors' rebuttal to this concern is just completely wrong. There is an entire literature using anti-CD25 or Treg-depleted mice (Foxp3-DTR) to acutely examine the effects of Treg depletion on disease progress. Phenotyping now shows that GITRL is expressed at an even higher level by MZ B cells, which neither augment Treg proliferation nor inhibit EAE – so regulation is not simply a matter of GITR expression. Finally, strictly speaking, the IL-10 independent nature of regulation in EAE by BDL per se was not shown (although the authors have previously shown that total B cells can act in an IL-10 independent manner in this uMT model – it should be shown that BDL, a subset of total B cells, also have the capacity to act in an IL-10 independent manner to prevent disease).

4. Concerns about generalizability of findings: This concern may not have been well enough stated in my initial review, and the authors' rebuttal focused on strain as opposed to model. Lets try again. The concern about generalizability of the findings relates to their being tested only in uMT mice, which exhibit a deficiency in Tregs (as well as defects in splenic architecture) and may be particularly susceptible to a Breg that can fix this underlying defect. BDL arguably work better in EAE than CHS (at least in terms of Treg proliferation see Fig 3A vs. 3G), a model where uMT mice have also undergone partial irradiation making them lymphopenic. This is not to question the ability of BDL to induce proliferation of Tregs in an IL-10 independent manner or to ameliorate EAE or CHS – but if these cells only work and through this mechanism in uMT mice, it alters the impact of the findings. The concern over the generalizability of findings is highlighted by a number of EAE studies where Bregs have been shown to act in an IL-10-dependent fashion in other EAE models (referenced in the initial review). These include more physiological settings such as mixed BM chimeras, which avoid uMT mice and/or cell transfer. Moreover, as noted previously, others have shown that B cells deficiency/depletion does not alter Treg number/function and/or that in EAE Bregs and Tregs operate independently. Therefore reassurance that BDL cells can act in a different setting is important in terms of understanding their overall role in the immune system. The CHS model is a nice control, but still utilizes uMT mice and was not used to demonstrate either IL-10-dependence or that Foxp3 expansion is necessary for regulation observed.

Other Concerns:

1. Fig 2A/I gating: Can the authors explain why their FO B gate and MZ gates include cells that are just as bright in CD93 as the main populations of T1/T2 cells? The expression of CD93 on the BDL cells is cut off right at its peak by the gating of the T3 population. Are the authors sure that the T3 population does not include BDL cells?

2. Fig 2A: While not a major part of the paper, the figure does not easily identify differences between numbers and frequency of cells in a given subset / gate. For example, actual effects of anti-CD20 on B cell depletion within each subset would be better identified by showing numbers of cells (as in Fig 1 for MZ and FOB) and not frequency of cells within gates derived from other gates. If proportion of cells is being assessed – it is important to also say proportion of what denominator? For example, amongst all remaining B cells after depletion, T1/T2 cells are reduced ~40%, and within the T1/T2 subset, T1 cells are increased by 25% and T2 cells are reduced by 30%. Because we already know that MZ B cells are not depleted (Fig 1a) we can surmise that the fall in % MZP cells in the MZ+ MZP gate actually reflects a fall in MZP numbers. The results are just not described accurately as they should be. It is also not clear whether these differences are specific to this representative example, or are reflected by other

mice. This is where showing cumulative data IN ADDITION to the dot plots is informative.

3. The data in Fig 3 are convincing that BDL supports Tregs. Interestingly the ratio of Tconv:Tregs is not increased by BDL, at least in the spleen. Do the authors have data on how BDL affect the encephalitogenic T cells? Do they also go up as do endogenous Tconv cells in the presence of BDL? Findings for Tregs and Teff in the dLN or CNS would be instructive. The statement "A reduction in Treg and conventional CD4 T cells on day 7 as compared to day 0 (Fig. 3A) is consistent with the mice being sublethally irradiated for EAE induction " (page 11 line 4-6) is inaccurate. Fig 3A does not show Tconv cells. As for Foxp3 cells, they are reduced in irradiated WT mice (3A vs. 3D, but they do not seem to be reduced in uMT mice on d7 despite irradiation.

4. Fig 7 shows that BDL can differentiate into FO and that both populations remain relatively stable in terms of percentage. The authors' interpretation is only one of several. Since the populations are defined solely by their IgD expression along a continuum, it is possible that in some cells IgD expression moves up changing their classification from BDL to FOB. These two populations may interconvert according to unknown cues or in a stochastic fashion, and only give the appearance of stable and converted populations. To prove the authors' interpretation one would need to retransfer BDL and FO cells and show they no longer "interconverted".

5. Fig 9: Importantly, to show that these human B cells are really the functional equivalent of the murine BDL subpopulation, the authors should show that anti-IL-10 does not influence the Treg proliferation, and that these cells express increased GITRL. Regarding gating, mature B cells express CD24 at lower levels than do transitional B cells but how was the cutoff for "CD24-" mature B cells determined (fig 9B)? Presumably, the "total" B cells used in 9D are within the CD19+CD20+CD24lo population that was not sorted for IgDlo?

Reviewer #3 (Remarks to the Author):

In general, I am satisfied by the authors responses; they have made appropriate changes to the manuscript.

The one response that I find very puzzling, is to my point #6, asking for data in patients treated with anti-CD20. They state that they have not done it yet because they "lack a definitive BDL cell surface phenotype in humans". This is curious, given the fact that they report data on these cells in humans in the paper, in the last part of the Results. If they don't have a phenotype, how can they report data on these cells?

Rebuttal

We thank reviewer #2 for such a thorough review that led to a better reporting of our novel findings. We have responded to all comments. Changes to the text are highlighted in yellow.

Reviewer #2

Major Concerns:

1. BDL are not distinct subset: While IgD^{lo} FO B cells are a subpopulation based on this phenotype and their function, they are not a “distinct subset” as claimed by the authors. The expression of IgD is a continuum and there is no distinct phenotypic subset as claimed on pg. 9. In their rebuttal, the authors note that other B cell subsets are not cleanly separated phenotypically. Nonetheless, markers used in combination allow most of the widely accepted B cell subsets to be separated and moreover, these also differ in their development, localization and chemotaxis within the SLO, activation/survival cues, and in some cases (plasma cells, GC B cells) transcription factors and signaling pathways (Notch2:MZ). Rather, BDL cells appear to be a subset of FO B cells that are at the end of IgD expression continuum that exhibit regulatory function. The cut-off between these cells and standard FO cells is arbitrary and with no reason to draw the gate in one place versus another (hence not distinct). If one examined the transcription profile of B cells just on either side of the cursor in Fig 2B, it is unlikely that they would give a dramatic step-off seen when the entire populations of FOB and BDL cells is assessed. The differences in transcriptional profile are primarily in genes related to proliferation and cell cycling and thus it cannot be claimed this population is a distinct subset (See below). This agrees with the noted increase in BDL homeostatic proliferation. New phenotyping in Fig S4 are not able to further differentiate these subsets because differences with FO B cells are marginal. Finally, the new data in Fig 7 and 8 suggest that some BDL can convert into and may be in equilibrium with FOB cells – again suggesting that BDL are not a distinct subset. Taken together these do not support – again not a distinct subset. By comparison, Kahn et al (Nat Com 2015) are extremely careful not to call their PD-L1 high B cells that regulate Tfh and humoral immunity a distinct subset. Bao et al (Cell Research, 2014) describe a phenotypically distinct population of CD11a/FCgRII^{hi} B cells with distinct transcription factor expression and are still reserved – calling it a new subset. The words distinct cannot be ascribed to this subset and data used to push for this designation are generally over-interpreted.

Author response: We are fine with defining BD_L as a new B cell subset. In retrospect using “new” instead of “distinct” is more in line with our results. The entire manuscript has been edited to reflect the change in verbiage.

2. Interpretation of transcriptome data: The differences in transcriptome identified by RNAseq as noted above, are in a single dimension and are primarily related to cell cycling and proliferation suggesting that this is a subset of FO cells that is activated and/or undergoing higher homeostatic proliferation. This reviewer does not agree with the author’s rebuttal that there is no B cell activation in naïve mice – they have GCs and have Plasma cells due to responses to environmental and endogenous antigens. REGARDLESS, B cells (especially BDL and transitional B cells) do undergo homeostatic proliferation in naïve mice and especially in lymphopenic hosts – as shown by the authors. Thus, BDL appears to be a subset of FO B cells undergoing proliferation and is possibly even auto-reactive.

Author response: BD_L could very well be a subset of FO B cells. We don’t favor that interpretation of our data, but at this point we cannot exclude it. That possibility is included in the Discussion on page 24. At this point, we cannot comment on whether BD_L are autoreactive, but because the FO subset is not considered autoreactive, if BD_L were that would actually be an argument against BD_L being a subset of FO B cells.

3. Mechanism is not proven: The regulatory activity of BDL is correlated with an increase in Tregs – but this mechanism is never proven. Moreover, BDL may have regulatory effects not related to Tregs. This could be

examined by depleting or blocking Tregs, for example with anti-CD25 (no breeding of new strains required). The authors' rebuttal to this concern is just completely wrong. There is an entire literature using anti-CD25 or Treg-depleted mice (Foxp3-DTR) to acutely examine the effects of Treg depletion on disease progress. Phenotyping now shows that GITRL is expressed at an even higher level by MZ B cells, which neither augment Treg proliferation nor inhibit EAE – so regulation is not simply a matter of GITR expression. Finally, strictly speaking, the IL-10 independent nature of regulation in EAE by BDL per se was not shown (although the authors have previously shown that total B cells can act in an IL-10 independent manner in this uMT model – it should be shown that BDL, a subset of total B cells, also have the capacity to act in an IL-10 independent manner to prevent disease).

Author response: As someone who has been in the field of B cell regulation from its beginnings, I (Dr. Dittel) am very perplexed with the laser focus on IL-10 regulatory mechanisms. We could provide many reasons for why much of the IL-10 Breg literature is not definitive. Please refer to our reviews on the topic. We have demonstrated that the mechanism of BDL_L regulation is not IL-10 (see Suppl. Fig. 2B) and performing the experiment again is not a good use of mice or laboratory resources. Even if IL-10 were involved it would not change the message of our manuscript, which is that we defined a new subset of B cells that play an important role in immune tolerance. We disagree that anti-CD25 is a good approach to deplete Treg since it has been shown to lead to functional inactivation of Treg and not their depletion (PMID: 16517695). The DTR approach is feasible, but again the expansion of Treg was just a readout to discern that BDL_L exhibit a unique function that could be separated from FO IgD^{hi} and MZ B cells. We never claimed that GITRL was the only mechanism, just that BDL_L require its expression to drive Treg expansion. The current study is focused on B cells, not Treg.

4. Concerns about generalizability of findings: This concern may not have been well enough stated in my initial review, and the authors' rebuttal focused on strain as opposed to model. Lets try again. The concern about generalizability of the findings relates to their being tested only in uMT mice, which exhibit a deficiency in Tregs (as well as defects in splenic architecture) and may be particularly susceptible to a Breg that can fix this underlying defect. BDL_L arguably work better in EAE than CHS (at least in terms of Treg proliferation see Fig 3A vs. 3G), a model where uMT mice have also undergone partial irradiation making them lymphopenic. This is not to question the ability of BDL_L to induce proliferation of Tregs in an IL-10 independent manner or to ameliorate EAE or CHS – but if these cells only work and through this mechanism in uMT mice, it alters the impact of the findings. The concern over the generalizability of findings is highlighted by a number of EAE studies where Bregs have been shown to act in an IL-10-dependent fashion in other EAE models (referenced in the initial review). These include more physiological settings such as mixed BM chimeras, which avoid uMT mice and/or cell transfer. Moreover, as noted previously, others have shown that B cell deficiency/depletion does not alter Treg number/function and/or that in EAE Bregs and Tregs operate independently. Therefore reassurance that BDL_L cells can act in a different setting is important in terms of understanding their overall role in the immune system. The CHS model is a nice control, but still utilizes uMT mice and was not used to demonstrate either IL-10-dependence or that Foxp3 expansion is necessary for regulation observed.

Author response: Everything the reviewer states regarding the disadvantages of using μ MT mice are correct. Every model as caveats and we used it as a tool as did the numerous studies demonstrating a role for IL-10 in B cell immune regulation. Again, we are perplexed with the laser focus on IL-10 dependence. As stated above, our study is about defining a new B cell subset with regulatory activity. It is not about Treg and their immune tolerance functions. The Treg side of the story is interesting and we would very much like to investigate that question, but that would comprise an entire manuscript of its own. We developed a number of new assays in which to demonstrate that BDL_L can be functionally distinguished from FO IgD^{hi} B cells. We disagree that other studies did not demonstrate that B cell depletion did not alter Treg number. Many studies actually report the data as percent Treg, not absolute number. If we reported our data as percent of CD4 T cells, we would also not see a difference. In regards to function, we also reported that B cell deficiency did not alter their function (Ray, 2012). In regards to BDL_L functioning similarly in a different environment/model we have had the same concern.

That is why we were very pleased to read the paper by Moore, 2017, who also observed B cell regulation via GITRL in Friend's virus infection.

Other Concerns:

1. Fig 2A/I gating: Can the authors explain why their FO B gate and MZ gates include cells that are just as bright in CD93 as the main populations of T1/T2 cells? The expression of CD93 on the BDL cells is cut off right at its peak by the gating of the T3 population. Are the authors sure that the T3 population does not include BDL cells?

Author response: We thank the reviewer for this observation. We cannot explain why in the representative example of our gating strategy the CD93 expression was high in the panel showing MZ and T2-MZP. That data was generated some time ago and thus we cannot go back and evaluate whether there was a problem with the antibodies. Thus we have replaced it with a more recent experiment that shows the identical enrichment for IgD^{low/-} B cells following 14 days of B cell depletion with anti-CD20 (Fig. 2B). The two experiments were conducted years apart by two co-first authors, thus we are confident in the finding. In the new representative example CD93 expression by FO and MZ B cells is what is expected. In addition, the same antibodies were used for Fig. 2A,B as for Fig. 2C-I, thus the new representative gating strategy is more appropriate to show. We cannot say for sure whether the T3 population contains some BDL. We have never tested that possibility because T3 have been reported to not give rise to mature B cells (PMID: 17548583). We are also intrigued by the CD93 expression in the BDL population. [Redacted]

2. Fig 2A: While not a major part of the paper, the figure does not easily identify differences between numbers and frequency of cells in a given subset / gate. For example, actual effects of anti-CD20 on B cell depletion within each subset would be better identified by showing numbers of cells (as in Fig 1 for MZ and FOB) and not frequency of cells within gates derived from other gates. If proportion of cells is being assessed – it is important to also say proportion of what denominator? For example, amongst all remaining B cells after depletion, T1/T2 cells are reduced ~40%, and within the T1/T2 subset, T1 cells are increased by 25% and T2 cells are reduced by 30%. Because we already know that MZ B cells are not depleted (Fig 1a) we can surmise that the fall in % MZP cells in the MZ+ MZP gate actually reflects a fall in MZP numbers. The results are just not described accurately as they should be. It is also not clear whether these differences are specific to this representative example, or are reflected by other mice. This is where showing cumulative data IN ADDITION to the dot plots is informative.

Author response: The purpose of Fig. 2A was just to show our gating strategy. It was not intended to depict the level of depletion in each B cell subset. The reviewer is correct regarding percentages and they were put on the panels just to give the reviewer a general idea of what B cells are and are not depleted. Thus we have removed the text describing the depletion of specific transitional subsets. The cumulative depletion data for FO and MZ B cells is shown in Fig. 1A.

3. The data in Fig 3 are convincing that BDL supports Tregs. Interestingly the ratio of Tconv:Tregs is not increased by BDL, at least in the spleen. Do the authors have data on how BDL affect the encephalitogenic T cells? Do they also go up as do endogenous Tconv cells in the presence of BDL? Findings for Tregs and Teff in the dLN or CNS would be instructive. The statement “A reduction in Treg and conventional CD4 T cells on day 7 as compared to day 0 (Fig. 3A) is consistent with the mice being sublethally irradiated for EAE induction ” (page 11 line 4-6) is inaccurate. Fig 3A does not show Tconv cells. As for Foxp3 cells, they are reduced in irradiated WT mice (3A vs. 3D, but they do not seem to be reduced in uMT mice on d7 despite irradiation.

Author response: We also find it interesting that the ratio of Tconv:Treg is not increased. Based on our and the studies of others there seems to be some unknown mechanism that keeps the ratio of Tconv:Treg consistent.

Just to speculate maybe that is why Treg adoptive immunotherapy does not lead to a sustained increase in Treg. We do not have any specific data on encephalitogenic T cells, although that is an interesting question and would be instructive, but is beyond the scope of the current study. The reviewer makes an interesting observation regarding Treg in nonirradiated (Fig. 3A) mice versus 7 days after irradiation (Fig. 3D) that we missed. We do not know why the Treg were not reduced, but could be due to a number of reasons. Since the above verbiage is not necessary to the interpretation of the results, we have removed it.

4. Fig 7 shows that BDL can differentiate into FO and that both populations remain relatively stable in terms of percentage. The authors' interpretation is only one of several. Since the populations are defined solely by their IgD expression along a continuum, it is possible that in some cells IgD expression moves up changing their classification from BDL to FOB. These two populations may interconvert according to unknown cues or in a stochastic fashion, and only give the appearance of stable and converted populations. To prove the authors' interpretation one would need to retransfer BDL and FO cells and show they no longer "interconverted".

Author response: We agree with the reviewer that our interpretation is only one of many. We provided several alternative interpretations in the Discussion. However, it is our prerogative as the authors to defend our favored hypothesis. It is up to the reader of the paper to agree or disagree with us. I (Dr. Dittel) have read many a paper in which I did not agree with the author's interpretation of their data. It is unclear to us how BDL and FO B cells would interconvert since we have never found FO B cells to induce Treg proliferation. In addition, we have never seen a study suggesting that FO and MZ B cells can interconvert. It seems that B cell maturation is an oneway street, but plasticity in B cell subsets like there is for Tconv/Treg and Th1/Th17 is a novel concept.

5. Fig 9: Importantly, to show that these human B cells are really the functional equivalent of the murine BDL subpopulation, the authors should show that anti-IL-10 does not influence the Treg proliferation, and that these cells express increased GITRL. Regarding gating, mature B cells express CD24 at lower levels than do transitional B cells but how was the cutoff for "CD24^{low}" mature B cells determined (fig 9B)? Presumably, the "total" B cells used in 9D are within the CD19⁺CD20⁺CD24^{lo} population that was not sorted for IgD^{lo}?

Author response: We are not sure why the reviewer is so focused on IL-10. What evidence is there that IL-10 induces Treg proliferation? It has been shown to play a role in Treg induction in humans when in the presence of TGFβ (PMID: 26363058). We do not add TGFβ to our in vitro culture system. We specifically measured proliferation, not Treg induction. The cut off for CD24 was based on CD24 expression levels in the CD20⁺ population in Fig. 9B. Total B cells are from the CD19 gate and would include IgD^{low/-} B cells. But if you do the calculation, IgD^{low/-} B cells represent only ~17% of the total B cell pool. Thus the number of B cells capable of inducing Treg expansion in the total B cell pool was likely too small to drive a measureable expansion in Treg.

Reviewer #3

In general, I am satisfied by the authors responses; they have made appropriate changes to the manuscript.

The one response that I find very puzzling, is to my point #6, asking for data in patients treated with anti-CD20. They state that they have not done it yet because they "lack a definitive BDL cell surface phenotype in humans". This is curious, given the fact that they report data on these cells in humans in the paper, in the last part of the Results. If they don't have a phenotype, how can they report data on these cells?

Author response: We have a phenotype that being IgD^{low/-}. But the phenotyping strategy that we used to identify mature B cells will include multiple subsets. For instance, IgD^{low/-} B cells will contain isotype class switched and memory B cells. All we could conclude is that regardless of the number of subsets contained with the IgD^{low/-} population a subset of B cells exists with the capacity to drive Treg expansion. We didn't claim anything over and above that.

REVIEWERS' COMMENTS:

Reviewer #2 (Remarks to the Author):

NCOMMS-18-08428, Ray, et al.

My comments in response to the authors' point by point reply are below their responses to my previous points.

Major Concerns:

1. BDL are not distinct subset:

Author response: We are fine with defining BDL as a new B cell subset. In retrospect using "new" instead of "distinct" is more in line with our results. The entire manuscript has been edited to reflect the change in verbiage.

Reviewer's response: The term "new subset" is acceptable.

2. Interpretation of transcriptome data: BDL appears to be a subset of FO B cells undergoing proliferation and is possibly even auto-reactive.

Author response: BDL could very well be a subset of FO B cells. We don't favor that interpretation of our data, but at this point we cannot exclude it. That possibility is included in the Discussion on page 24. At this point, we cannot comment on whether BDL are autoreactive, but because the FO subset is not considered autoreactive, if BDL were that would actually be an argument against BDL being a subset of FO B cells.

Reviewer's response: Regardless of what you want to call it, B cells undergo homeostatic proliferation and the small proportion of B cells with FO phenotype at the tail end of the continuum of IgD expression may exhibit more homeostatic proliferation than the rest of FO B cells and/or be more responsive to BAFF (borne out in Fig 7). The bottom line: the transcriptomic data do not support that these cells are a distinct subset – as discussed previously. I agree with changing the verbiage, as above, to a "new subset".

3. Mechanism is not proven:

Author response: As someone who has been in the field of B cell regulation from its beginnings, I (Dr. Dittel) am very perplexed with the laser focus on IL-10 regulatory mechanisms. We could provide many reasons for why much of the IL-10 Breg literature is not definitive. Please refer to our reviews on the topic. We have demonstrated that the mechanism of BDL regulation is not IL-10 (see Suppl. Fig. 2B) and performing the experiment again is not a good use of mice or laboratory resources. Even if IL-10 were involved it would not change the message of our manuscript, which is that we defined a new subset of B cells that play an important role in immune tolerance. We disagree that anti-CD25 is a good approach to deplete Treg since it has been shown to lead to functional inactivation of Treg and not their depletion (PMID: 16517695). The DTR approach is feasible, but again the expansion of Treg was just a readout to discern that BDL exhibit a unique function that could be separated from FO IgDhi and MZ B cells. We never claimed that GITRL was the only mechanism, just that BDL require its expression to drive Treg expansion. The current study is focused on B cells, not Treg.

Reviewer's response: the authors are trying to deflect this reviewer's argument, which is, the mechanism of action for BDL is based on association and more definitive studies are necessary. FIRST: This manuscript extends a previous study by the authors identifying Bregs that inhibit EAE by expanding Tregs in a GITR-L dependent and IL-10 independent manner. The authors have discovered

a new subset of B cells that exhibit some of the same regulatory properties as they previously described for the whole B cell population. It is now important to show they really work the same way, or determine how they are different. Second: To increase novelty of this new manuscript enough to now justify a higher impact publication than in their earlier work, they need to provide more definitive evidence of the mechanism of action (especially since this manuscript does not show that BDL are a distinct subset). Third: The author's appear to want it "both ways". In the abstract, they claim that BDL "... maintain tolerance by promoting CD4+Foxp3+ regulatory T cell (Treg) homeostatic expansion". This is not proven. On pg 9, the subheading and introduction to experiments in Fig 3 state: "BDL B cells ... exhibit B cell regulatory activity in an IL-10-independent manner" and "To determine whether BDL specifically exhibit B cell regulatory activity, uMT mice were reconstituted ... and Treg numbers were quantitated 10 days later." So clearly the authors are claiming mechanism of action in an IL-10 independent and Treg dependent manner. Now they need to go past the previous studies and provide more definitive evidence of the mechanism(s) of action. While the increase in Tregs and Treg proliferation are associated with improved EAE this reviewer has asked the authors to provide more mechanistic insights into the mechanism of action of this "new subset" of B cells. For example, the role of GITR expression by BDL is linked only to its requirement for Treg proliferation and number, but not to EAE disease. The only experiment shown for IL-10-dependence is that BDL support Treg proliferation in an IL-10 independent manner. However: 1) While GITR-L expression may be required for Treg proliferation, it is NOT the mechanism of action of BDL cells, since MZ B cells express significantly more GITRL than BDL (Fig 5) and yet MZ B cells are neither suppressive in this model, nor do they expand Tregs. (This point should be mentioned by the authors on page 14). So in summary, GITR-L is required for Treg proliferation but does not explain it. In fact the mechanism of increased Treg proliferation is unknown. 2) WE do not know if BDL actually inhibit EAE in an IL-10 independent manner – as they claim. The mention of IL-10 is brought up by this reviewer and by the authors themselves because IL-10 dependence has been demonstrated in EAE by several different groups who are also well-established in the Breg field. In this regard, others have shown that their IL-10-dependent Bregs ALSO induce Tregs. Moreover the authors have not excluded that BDL might not also augment Tregs through enhanced conversion – in an IL-10 dependent manner. Finally, it has been clearly shown that IL-10 signaling confers Tregs with improved ability to control IL-17 responses (Rundensky 2011), which is clearly involved in EAE pathogenesis. So BDL may utilize more than one mechanism and the mechanistic studies on BDL are incomplete. We know less about how they work than the whole B cells previously studied by the authors. The studies requested are relatively easily performed and the reviewer is concerned about the authors' reluctance to shore up their mechanistic data.

4. Concerns about generalizability of findings:

Author response: Everything the reviewer states regarding the disadvantages of using μ MT mice are correct. Every model as caveats and we used it as a tool as did the numerous studies demonstrating a role for IL-10 in B cell immune regulation. Again, we are perplexed with the laser focus on IL-10 dependence. As stated above, our study is about defining a new B cell subset with regulatory activity. It is not about Treg and their immune tolerance functions. The Treg side of the story is interesting and we would very much like to investigate that question, but that would comprise an entire manuscript of its own. We developed a number of new assays in which to demonstrate that BDL can be functionally distinguished from FO IgDhi B cells. We disagree that other studies did not demonstrate that B cell depletion did not alter Treg number. Many studies actually report the data as percent Treg, not absolute number. If we reported our data as percent of CD4 T cells, we would also not see a difference. In regards to function, we also reported that B cell deficiency did not alter their function (Ray, 2012). In regards to BDL functioning similarly in a different environment/model we have had the same concern.

Reviewer's response: the authors again deflect the argument. Of course every model has its limitations. The point is that to publish their second manuscript on these Bregs in a higher impact journal the authors need to go further and show that their findings are not only seen in transfer models into uMT mice. (Others find different Breg mechanisms in non-transfer / non-lymphopenic models of EAE). Finding a regulatory subset that acts in a unique manner in one specific type of model carry less impact, especially to immunologists who are not focused on Bregs, than regulatory cells that have more generalizable activity. Regarding mechanistic studies see my comments above.

Other Concerns:

1. Fig 2A/I gating:

Author response: We thank the reviewer for this observation. We cannot explain why in the representative example of our gating strategy the CD93 expression was high in the panel showing MZ and T2-MZP. That data was generated some time ago and thus we cannot go back and evaluate whether there was a problem with the antibodies. Thus we have replaced it with a more recent experiment that shows the identical enrichment for IgDlow/- B cells following 14 days of B cell depletion with anti-CD20 (Fig. 2B). The two experiments were conducted years apart by two co-first authors, thus we are confident in the finding. In the new representative example CD93 expression by FO and MZ B cells is what is expected. In addition, the same antibodies were used for Fig. 2A,B as for Fig. 2C-I, thus the new representative gating strategy is more appropriate to show. We cannot say for sure whether the T3 population contains some BDL. We have never tested that possibility because T3 have been reported to not give rise to mature B cells (PMID: 17548583). We are also intrigued by the CD93 expression in the BDL population. **[Redacted]**

Reviewer's response: New representative figure is improved.

It is premature to propose a new transitional subset when the data supporting that BDL are a distinct lineage from FO B cells are so completely debatable. See point 4 below.

2. Fig 2A: does not easily identify differences between numbers and frequency of cells in a given subset / gate.

Author response: The purpose of Fig. 2A was just to show our gating strategy. It was not intended to depict the level of depletion in each B cell subset. The reviewer is correct regarding percentages and they were put on the panels just to give the reviewer a general idea of what B cells are and are not depleted. Thus we have removed the text describing the depletion of specific transitional subsets. The cumulative depletion data for FO and MZ

B cells is shown in Fig. 1A.

Reviewer's response: acceptable

3. The data in Fig 3 are convincing that BDL supports Tregs. Interestingly the ratio of Tconv:Tregs is not increased by BDL, at least in the spleen.

Author response: We also find it interesting that the ratio of Tconv:Treg is not increased. Based on our and the studies of others there seems to be some unknown mechanism that keeps the ratio of Tconv:Treg consistent.

Just to speculate maybe that is why Treg adoptive immunotherapy does not lead to a sustained

increase in Treg. We do not have any specific data on encephalitogenic T cells, although that is an interesting question and would be instructive, but is beyond the scope of the current study. The reviewer makes an interesting observation regarding Treg in nonirradiated (Fig. 3A) mice versus 7 days after irradiation (Fig. 3D) that we missed. We do not know why the Treg were not reduced, but could be due to a number of reasons. Since the above verbiage is not necessary to the interpretation of the results, we have removed it.

Reviewer's response: in most studies on Tregs, the sine qua non for regulation is to boost Tregs out of proportion to Tconv cells. Here BDL support Tregs but equally support Tconv. This goes back to lack of proof of this as the mechanism by which BDL act. Since the CNS cells may differ, it was hoped that the authors might examine whether the Treg response was higher than the Tconv response at this site, adding evidence to buttress their mechanistic claims.

4. Fig 7 shows that BDL can differentiate into FO and that both populations remain relatively stable in terms of percentage. The authors' interpretation is only one of several. Since the populations are defined solely by their IgD expression along a continuum, it is possible that in some cells IgD expression moves up changing their classification from BDL to FOB. These two populations may interconvert according to unknown cues or in a stochastic fashion, and only give the appearance of stable and converted populations. To prove the authors' interpretation one would need to retransfer BDL and FO cells and show they no longer "interconverted".

Author response: We agree with the reviewer that our interpretation is only one of many. We provided several alternative interpretations in the Discussion. However, it is our prerogative as the authors to defend our favored hypothesis. It is up to the reader of the paper to agree or disagree with us. I (Dr. Dittel) have read many a paper in which I did not agree with the author's interpretation of their data. It is unclear to us how BDL and FO B cells would interconvert since we have never found FO B cells to induce Treg proliferation. In addition, we have never seen a study suggesting that FO and MZ B cells can interconvert. It seems that B cell maturation is an oneway street, but plasticity in B cell subsets like there is for Tconv/Treg and Th1/Th17 is a novel concept.

Reviewer's response: It is the MAJOR role of reviewers to make sure that the interpretations are supported by the data and that the authors present a balanced view.

85% of BDL convert into FO B cells within one day. The fraction remaining as BDL (12%) remains constant between day 1 and day 2. BDL appear to exhibit increased homeostatic proliferation and the question remains whether this is simply a subpopulation of FO B cells that has BCRs that make them more reactive to auto-, environmental, or microbiota-related Ags when transferred into a new host, or whether they are more responsive to BAFF. It is completely unknown whether they will remain distinct once lymphopenia resolves, or on re-transfer of the remaining BDL cells to a second host. Thus the conclusion (page 16) stating that "BDL likely contains two new B cell subsets one that differentiates into FO B cells and the other the mature BDL subset." While this may be possible or suggestive, "likely" is premature without further studies. With due respect, Dr. Dittel has no idea whether or not BDL can interconvert into FO B cells and vice versa, because the BDL subset has just been identified and there is scant evidence that it is truly a distinct subset. The comparison to T cell subsets defined by transcription factors is quite a leap in faith.

Additional question: 12% of transferred BDL remained IgDlo in the one representative experiment. What was this percentage in the other three experiments? Was this consistent? It is preferable to ALSO show average and SD rather than ONLY "representative" data.

5. Fig 9: Importantly, to show that these human B cells are really the functional equivalent of the murine BDL subpopulation, the authors should show that anti-IL-10 does not influence the Treg proliferation, and that these cells express increased GITRL.

Author response: We are not sure why the reviewer is so focused on IL-10. What evidence is there that IL-10 induces Treg proliferation? It has been shown to play a role in Treg induction in humans

when in the presence of TGFb (PMID: 26363058). We do not add TGFb to our in vitro culture system. We specifically measured proliferation, not Treg induction. The cut off for CD24 was based on CD24 expression levels in the CD20- population in Fig. 9B. Total B cells are from the CD19 gate and would include IgDlow/- B cells. But if you do the calculation, IgDlow/- B cells represent only ~17% of the total B cell pool. Thus the number of B cells capable of inducing Treg expansion in the total B cell pool was likely too small to drive a measureable expansion in Treg.

Reviewer's response: again the authors try to deflect from the reviewer's intent. They have identified a cell population in humans that appears to be the same as their murine population based on IgD. To show that it is really the SAME they should provide more detail about its phenotype and mechanisms of action.

It seems trivial to examine GITR-L expression on this subset, which must be done since this is the basis for their mechanistic claim for their murine BDL.

In addition:

A) Compared to the murine population, the IgDlow population of human splenic B cells is much larger and quite distinct from the IgDhi cells. Was the same seen in human PBLs? This should at least be mentioned.

B) In the proliferation experiments using soluble CD3 and CD28, it is likely that the APCs/B cells are providing FcR-mediated CD3-cross-linking required for optimal T cell proliferation. Especially in experiments APCs are absent, it is important to show levels of FcR on the IgDLo vs. IgD High B cells

C) In two places on page 17, the authors mistakenly refer to panels of Fig 9 as Fig 10.

6) New minor point:

1 Supplementary Fig 1 needs statistics shown.

Reviewer #3 (Remarks to the Author):

I appreciate the authors comments. I suspect that the reason to not look at rituximab treated patients is the extra work required, not the reason the authors give. That said, I am sympathetic and would not hold up publication by asking them to undertake that new study.

NCOMMS-18-08428, Ray, et al.
Point-by-Point Rebuttal

We have responded to all comments. Changes that are in response to reviewer or editorial requested changes are highlighted in yellow. Changes required to shorten the length of the manuscript are not highlighted.

Reviewer #2

1. BDL are not distinct subset:

Author response: We are fine with defining BDL as a new B cell subset. In retrospect using “new” instead of “distinct” is more in line with our results. The entire manuscript has been edited to reflect the change in verbiage.

Reviewer’s response: The term “new subset” is acceptable.

2. Interpretation of transcriptome data: BDL appears to be a subset of FO B cells undergoing proliferation and is possibly even auto-reactive.

Author response: BDL could very well be a subset of FO B cells. We don’t favor that interpretation of our data, but at this point we cannot exclude it. That possibility is included in the Discussion on page 24. At this point, we cannot comment on whether BDL are autoreactive, but because the FO subset is not considered autoreactive, if BDL were that would actually be an argument against BDL being a subset of FO B cells.

Reviewer’s response: Regardless of what you want to call it, B cells undergo homeostatic proliferation and the small proportion of B cells with FO phenotype at the tail end of the continuum of IgD expression may exhibit more homeostatic proliferation than the rest of FO B cells and/or be more responsive to BAFF (borne out in Fig 7). The bottom line: the transcriptomic data do not support that these cells are a distinct subset – as discussed previously. I agree with changing the verbiage, as above, to a “new subset”.

Author Response to points 1 and 2: We are pleased that we have come to an agreement as to how to define BDL.

3. Mechanism is not proven:

Author response: As someone who has been in the field of B cell regulation from its beginnings, I (Dr. Dittel) am very perplexed with the laser focus on IL-10 regulatory mechanisms. We could provide many reasons for why much of the IL-10 Breg literature is not definitive. Please refer to our reviews on the topic. We have demonstrated that the mechanism of BDL regulation is not IL-10 (see Suppl. Fig. 2B) and performing the experiment again is not a good use of mice or laboratory resources. Even if IL-10 were involved it would not change the message of our manuscript, which is that we defined a new subset of B cells that play an important role in immune tolerance. We disagree that anti-CD25 is a good approach to deplete Treg since it has been shown to lead to functional inactivation of Treg and not their depletion (PMID: 16517695). The DTR approach is feasible, but again the expansion of Treg was just a readout to discern that BDL exhibit a unique function that could be separated from FO IgDhi and MZ B cells. We never

claimed that GITRL was the only mechanism, just that BDL require its expression to drive Treg expansion. The current study is focused on B cells, not Treg.

Reviewer's response: the authors are trying to deflect this reviewer's argument, which is, the mechanism of action for BDL is based on association and more definitive studies are necessary. FIRST: This manuscript extends a previous study by the authors identifying Bregs that inhibit EAE by expanding Tregs in a GITR-L dependent and IL-10 independent manner. The authors have discovered a new subset of B cells that exhibit some of the same regulatory properties as they previously described for the whole B cell population. It is now important to show they really work the same way, or determine how they are different. Second: To increase novelty of this new manuscript enough to now justify a higher impact publication than in their earlier work, they need to provide more definitive evidence of the mechanism of action (especially since this manuscript does not show that BDL are a distinct subset). Third: The author's appear to want it "both ways". In the abstract, they claim that BDL "... maintain tolerance by promoting CD4+Foxp3+ regulatory T cell (Treg) homeostatic expansion". This is not proven. On pg 9, the subheading and introduction to experiments in Fig 3 state: "BDL B cells ... exhibit B cell regulatory activity in an IL-10-independent manner" and "To determine whether BDL specifically exhibit B cell regulatory activity, uMT mice were reconstituted ... and Treg numbers were quantitated 10 days later." So clearly the authors are claiming mechanism of action in an IL-10 independent and Treg dependent manner. Now they need to go past the previous studies and provide more definitive evidence of the mechanism(s) of action. While the increase in Tregs and Treg proliferation are associated with improved EAE this reviewer has asked the authors to provide more mechanistic insights into the mechanism of action of this "new subset" of B cells. For example, the role of GITR expression by BDL is linked only to its requirement for Treg proliferation and number, but not to EAE disease. The only experiment shown for IL-10-dependence is that BDL support Treg proliferation in an IL-10 independent manner. However: 1) While GITR-L expression may be required for Treg proliferation, it is NOT the mechanism of action of BDL cells, since MZ B cells express significantly more GITRL than BDL (Fig 5) and yet MZ B cells are neither suppressive in this model, nor do they expand Tregs. (This point should be mentioned by the authors on page 14). So in summary, GITR-L is required for Treg proliferation but does not explain it. In fact the mechanism of increased Treg proliferation is unknown. 2) WE do not know if BDL actually inhibit EAE in an IL-10 independent manner – as they claim. The mention of IL-10 is brought up by this reviewer and by the authors themselves because IL-10 dependence has been demonstrated in EAE by several different groups who are also well-established in the Breg field. In this regard, others have shown that their IL-10-dependent Bregs ALSO induce Tregs. Moreover the authors have not excluded that BDL might not also augment Tregs through enhanced conversion – in an IL-10 dependent manner. Finally, it has been clearly shown that IL-10 signaling confers Tregs with improved ability to control IL-17 responses (Rundensky 2011), which is clearly involved in EAE pathogenesis. So BDL may utilize more than one mechanism and the mechanistic studies on BDL are incomplete. We know less about how they work than the whole B cells previously studied by the authors. The studies requested are relatively easily performed and the reviewer is concerned about the authors' reluctance to shore up their mechanistic data.

Author response: We demonstrated that BDL induce Treg expansion in a GITRL-dependent manner in Fig. 5 using both antibody blocking and GITRL-deficient mice. We also showed that

they do so in an IL-10-independent manner. The abstract has been changed by the editors to more closely reflect the findings in our study including wording that BD_L, at least in part, promote Treg expansion in a GITRL-dependent manner. We are working on other mechanisms required for BD_L induction of Treg proliferation in addition to GITRL, but those studies are extensive and will comprise a separate manuscript. There are no IL-10-dependent Breg. There are only B cells that regulate in an IL-10-dependent manner, which includes a number of B cell subsets. IL-10 is a functional phenotype that is not specific to a particular B cell subset. It is possible that BD_L in addition to driving Treg proliferation also induce Treg from conventional CD4 T cells. We have never claimed that they do not. We plan to study the Treg side of the story in the future.

4. Concerns about generalizability of findings:

Author response: Everything the reviewer states regarding the disadvantages of using μ MT mice are correct. Every model as caveats and we used it as a tool as did the numerous studies demonstrating a role for IL-10 in B cell immune regulation. Again, we are perplexed with the laser focus on IL-10 dependence. As stated above, our study is about defining a new B cell subset with regulatory activity. It is not about Treg and their immune tolerance functions. The Treg side of the story is interesting and we would very much like to investigate that question, but that would comprise an entire manuscript of its own. We developed a number of new assays in which to demonstrate that BDL can be functionally distinguished from FO IgDhi B cells. We disagree that other studies did not demonstrate that B cell depletion did not alter Treg number. Many studies actually report the data as percent Treg, not absolute number. If we reported our data as percent of CD4 T cells, we would also not see a difference. In regards to function, we also reported that B cell deficiency did not alter their function (Ray, 2012). In regards to BDL functioning similarly in a different environment/model we have had the same concern.

Reviewer's response: the authors again deflect the argument. Of course every model has its limitations. The point is that to publish their second manuscript on these Bregs in a higher impact journal the authors need to go further and show that their findings are not only seen in transfer models into μ MT mice. (Others find different Breg mechanisms in non-transfer / non-lymphopenic models of EAE). Finding a regulatory subset that acts in a unique manner in one specific type of model carry less impact, especially to immunologists who are not focused on Bregs, than regulatory cells that have more generalizeable activity. Regarding mechanistic studies see my comments above.

Author response: The point about using only μ MT mice was a concern of ours as well. That is why we utilized B cell depletion strategies. In our previous study, we showed that total B cell depletion lead to a decrease in Treg and an inability to recover from EAE. Whereas, in this manuscript, we show that partial B cell depletion retains Treg and the mice are able to recover from EAE. In addition, we showed that partial B cell depletion did not alter the regulatory function of BD_L or lead to it.

Other Concerns:

1. Fig 2A/I gating:

Author response: We thank the reviewer for this observation. We cannot explain why in the

representative example of our gating strategy the CD93 expression was high in the panel showing MZ and T2-MZP. That data was generated some time ago and thus we cannot go back and evaluate whether there was a problem with the antibodies. Thus we have replaced it with a more recent experiment that shows the identical enrichment for IgDlow/- B cells following 14 days of B cell depletion with anti-CD20 (Fig. 2B). The two experiments were conducted years apart by two co-first authors, thus we are confident in the finding. In the new representative example CD93 expression by FO and MZ B cells is what is expected. In addition, the same antibodies were used for Fig. 2A,B as for Fig. 2C-I, thus the new representative gating strategy is more appropriate to show. We cannot say for sure whether the T3 population contains some BDL. We have never tested that possibility because T3 have been reported to not give rise to mature B cells (PMID: 17548583). We are also intrigued by the CD93 expression in the BDL population. As outlined in the Discussion, we have proposed that the BDL phenotype contains two populations, one of which is a new transitional subset, which we named T2-FOP. If that population exists, they would likely have higher levels of CD93 expression, which would explain our data. But, proof of that is well beyond the scope of the current manuscript.

Reviewer's response: New representative figure is improved.

It is premature to propose a new transitional subset when the data supporting that BDL are a distinct lineage from FO B cells are so completely debatable. See point 4 below.

Author response: [Redacted]

2. Fig 2A: does not easily identify differences between numbers and frequency of cells in a given subset / gate.

Author response: The purpose of Fig. 2A was just to show our gating strategy. It was not intended to depict the level of depletion in each B cell subset. The reviewer is correct regarding percentages and they were put on the panels just to give the reviewer a general idea of what B cells are and are not depleted. Thus we have removed the text describing the depletion of specific transitional subsets. The cumulative depletion data for FO and MZ B cells is shown in Fig. 1A.

Reviewer's response: acceptable

Author response: We are happy that the reviewer found the new data acceptable.

3. The data in Fig 3 are convincing that BDL supports Tregs. Interestingly the ratio of Tconv:Treg is not increased by BDL, at least in the spleen.

Author response: We also find it interesting that the ratio of Tconv:Treg is not increased. Based on our and the studies of others there seems to be some unknown mechanism that keeps the ratio of Tconv:Treg consistent. Just to speculate maybe that is why Treg adoptive immunotherapy does not lead to a sustained increase in Treg. We do not have any specific data on encephalitogenic T cells, although that is an interesting question and would be instructive, but is beyond the scope of the current study. The reviewer makes an interesting observation regarding Treg in nonirradiated (Fig. 3A) mice versus 7 days after irradiation (Fig. 3D) that we missed. We

do not know why the Treg were not reduced, but could be due to a number of reasons. Since the above verbiage is not necessary to the interpretation of the results, we have removed it.

Reviewer's response: in most studies on Tregs, the sine qua non for regulation is to boost Tregs out of proportion to Tconv cells. Here BDL support Tregs but equally support Tconv. This goes back to lack of proof of this as the mechanism by which BDL act. Since the CNS cells may differ, it was hoped that the authors might examine whether the Treg response was higher than the Tconv response at this site, adding evidence to buttress their mechanistic claims.

Author response: In this study, we did not examine any immune cells in the CNS. The data on the functionality of Treg within the CNS during EAE is controversial and has not been resolved to our knowledge. Thus looking at Treg numbers in the CNS would not provide any mechanistic insight into their function during EAE.

4. Fig 7 shows that BDL can differentiate into FO and that both populations remain relatively stable in terms of percentage. The authors' interpretation is only one of several. Since the populations are defined solely by their IgD expression along a continuum, it is possible that in some cells IgD expression moves up changing their classification from BDL to FOB. These two populations may interconvert according to unknown cues or in a stochastic fashion, and only give the appearance of stable and converted populations. To prove the authors' interpretation one would need to retransfer BDL and FO cells and show they no longer "interconverted".

Author response: We agree with the reviewer that our interpretation is only one of many. We provided several alternative interpretations in the Discussion. However, it is our prerogative as the authors to defend our favored hypothesis. It is up to the reader of the paper to agree or disagree with us. I (Dr. Dittel) have read many a paper in which I did not agree with the author's interpretation of their data. It is unclear to us how BDL and FO B cells would interconvert since we have never found FO B cells to induce Treg proliferation. In addition, we have never seen a study suggesting that FO and MZ B cells can interconvert. It seems that B cell maturation is an oneway street, but plasticity in B cell subsets like there is for Tconv/Treg and Th1/Th17 is a novel concept.

Reviewer's response: It is the MAJOR role of reviewers to make sure that the interpretations are supported by the data and that the authors present a balanced view.

85% of BDL convert into FO B cells within one day. The fraction remaining as BDL (12%) remains constant between day 1 and day 2. BDL appear to exhibit increased homeostatic proliferation and the question remains whether this is simply a subpopulation of FO B cells that has BCRs that make them more reactive to auto-, environmental, or microbiota-related Ags when transferred into a new host, or whether they are more responsive to BAFF. It is completely unknown whether they will remain distinct once lymphopenia resolves, or on re-transfer of the remaining BDL cells to a second host. Thus the conclusion (page 16) stating that "BDL likely contains two new B cell subsets one that differentiates into FO B cells and the other the mature BDL subset." While this may be possible or suggestive, "likely" is premature without further studies. With due respect, Dr. Dittel has no idea whether or not BDL can interconvert into FO B cells and vice versa, because the BDL subset has just been identified and there is scant evidence that it is truly a distinct subset. The comparison to T cell subsets defined by transcription factors is quite a leap in faith.

Additional question: 12% of transferred BDL remained IgDlo in the one representative

experiment. What was this percentage in the other three experiments? Was this consistent? It is preferable to ALSO show average and SD rather than ONLY “representative” data.

Author response: We do not know whether BD_L and FO interconvert, but what we do know is that IgD^{hi} B cells never induce Treg proliferation when isolated from unmanipulated mice. As discussed above, we have removed our speculation that an additional transitional B cell subset exists. We are currently working on proving that hypothesis. In addition, we have preliminary evidence that IgD expression is regulated in BD_L utilizing a mechanism that has not been previously described. If we are correct, then gain of IgD expression making BD_L phenotypically similar to FO would not necessarily mean that they have changed functionality. We have never claimed that IgD expression levels has anything to do with BD_L function. During EAE, the BD_L IgD^{low/-} phenotype was quite stable following adoptive transfer (Fig. 8).

5. Fig 9: Importantly, to show that these human B cells are really the functional equivalent of the murine BDL subpopulation, the authors should show that anti-IL-10 does not influence the Treg proliferation, and that these cells express increased GITRL.

Author response: We are not sure why the reviewer is so focused on IL-10. What evidence is there that IL-10 induces Treg proliferation? It has been shown to play a role in Treg induction in humans when in the presence of TGFb (PMID: 26363058). We do not add TGFb to our in vitro culture system. We specifically measured proliferation, not Treg induction. The cut off for CD24 was based on CD24 expression levels in the CD20- population in Fig. 9B. Total B cells are from the CD19 gate and would include IgD^{low/-} B cells. But if you do the calculation, IgD^{low/-} B cells represent only ~17% of the total B cell pool. Thus the number of B cells capable of inducing Treg expansion in the total B cell pool was likely too small to drive a measureable expansion in Treg.

Reviewer’s response: again the authors try to deflect from the reviewer’s intent. They have identified a cell population in humans that appears to be the same as their murine population based on IgD. To show that it is really the SAME they should provide more detail about its phenotype and mechanisms of action. It seems trivial to examine GITR-L expression on this subset, which must be done since this is the basis for their mechanistic claim for their murine BDL.

Author response: Measuring GITRL expression on a mixed population of B cells won’t demonstrate that the human BD_L equivalent does or does not express higher levels of GITRL than other B cell subsets since we cannot yet fully phenotype them. We are working on this, but don’t yet have sufficient additional markers.

In addition:

A) Compared to the murine population, the IgD^{low} population of human splenic B cells is much larger and quite distinct from the IgD^{hi} cells. Was the same seen in human PBLs? This should at least be mentioned.

Author response: In Suppl. Fig. 7, we show gating for IgD^{low/-} and IgD^{hi} peripheral blood B cells. It more closely resembles the mouse splenic data than the human spleen. We speculate it is

because in the human class switched memory B cells are not circulating and are retained within the spleen. Our mice under SPF conditions would have very few memory B cells.

B) In the proliferation experiments using soluble CD3 and CD28, it is likely that the APCs/B cells are providing FcR-mediated CD3-cross-linking required for optimal T cell proliferation. Especially in experiments APCs are absent, it is important to show levels of FcR on the IgD^{Lo} vs. IgD^{High} B cells

Author response: The FcR expressed by human B cells are FcγRIIa and FcγRIIb. These are low affinity inhibitory receptors that largely are engaged by immune complexes. Thus we don't see how monomeric antibody bound to B cells, which doesn't readily occur, could play a role in our assay. In the peripheral blood experiments (Fig. 9F) the antibodies were plate bound. In reading over the Methods, we found that that information was omitted leading to the confusion. We have now added the plate bound information into the Methods and we thank the reviewer for causing us to correct the omission.

C) In two places on page 17, the authors mistakenly refer to panels of Fig 9 as Fig 10.

Author response: Thank you for catching that and it has been corrected.

6) New minor point:

1 Supplementary Fig 1 needs statistics shown.

Author response: The statistics have been added.

Reviewer #3

I appreciate the authors comments. I suspect that the reason to not look at rituximab treated patients is the extra work required, not the reason the authors give. That said, I am sympathetic and would not hold up publication by asking them to undertake that new study.

Author response: We thank the reviewer for allowing our publication to proceed. You are correct in that the examination of B cells following rituximab would be a substantial amount of work. We now have a local collaborator/neurologist that is interested in phenotyping both B cells and Treg following B cell depletion. We are currently in the process of submitting grants to obtain the required funding.